# A Formal Comparison Between Chain-of-Thought and Latent Thought

## Abstract

Chain-of-Thought (CoT) elicits reasoning in large language models by explicitly generating intermediate steps in natural language. In contrast, Latent Thought in looped models operates directly in the continuous latent space, enabling computation beyond discrete linguistic representations. While both approaches exploit iterative computation, their comparative capabilities remain underexplored. In this work, we present a formal analysis showing that Latent Thought in looped Transformers enables parallel computation, which is more efficient than the inherently sequential process of CoT. In contrast, CoT leverages stochastic decoding to approximate solutions to problems where exact computation is intractable. These separations suggest the tasks for which depth-driven recursion is more suitable, thereby offering practical guidance for choosing between reasoning paradigms.

## 1 Introduction

Transformer (Vaswani et al., 2017)-based large language models (LLMs) have shown strong performance across diverse tasks and have recently been extended to complex reasoning. Rather than directly predicting final answers, prompting LLMs to generate intermediate reasoning steps, known as *Chain-of-Thought (CoT)* (Wei et al., 2022), has been shown to enhance reasoning capabilities. This naturally raises the question: *why is CoT effective for complex tasks?* Recent studies have approached this question by framing reasoning as a computational problem and analyzing its complexity Feng et al. (2023); Merrill & Sabharwal (2024); Li et al. (2024); Nowak et al. (2024), showing that CoT improves performance by increasing the model's *effective depth* through iterative computation, thereby enabling the solution of problems that would otherwise be infeasible.

Recent studies have proposed to enhance the expressive power of Transformer through *Latent Thought reasoning*, which operates in the hidden state space rather than the linguistic space, with the potential to achieve more expressive and efficient reasoning (Hao et al., 2025; Saunshi et al., 2025). One prominent approach is *looped Transformer (looped TF)*, where Transformer blocks are applied recurrently by feeding outputs back as inputs (Dehghani et al., 2019). Such iterative structures have been shown to extend expressivity (Giannou et al., 2023; Xu & Sato, 2025). In addition, they achieve competitive performance with fewer parameters (Lan et al., 2020; Csordás et al., 2024; Bae et al., 2025) , improve performance on complex reasoning tasks (Saunshi et al., 2025; Wang et al., 2025).

These reasoning approaches share the common idea of iteratively applying Transformers to enhance expressive power, which naturally raises a fundamental question:

*What is the separation between Chain-of-Thought and Latent Thought (looped Transformer)?*

Motivated by this question, recent studies have compared their expressive power under comparable numbers of iterations. Saunshi et al. (2025) showed that deterministic CoT is contained within looped Transformers by observing that the decoding step in CoT can be made deterministic. In addition, Merrill & Sabharwal (2025a) demonstrated a separation in the logarithmic regime by characterizing the expressive power of these models through computational complexity theory. While these results suggest that looped models can be more expressive than CoT, several fundamental questions remain:

*Does this separation extend beyond the logarithmic regime?*
*Are looped Transformers always more expressive than CoT?*

## 1.1 OUR CONTRIBUTIONS

We address these questions by revealing the distinct strengths and limitations of each paradigm under different task settings. Specifically, we show that Latent Thought exhibits clear advantages in the polylogarithmic regime, whereas this superiority does not necessarily hold universally, there exist scenarios where chain-of-thought reasoning performs better. A summary of is provided in Figure 4.

**Latent Thought Enables Efficient Reasoning**  For many reasoning tasks, not only the correctness of a solution but also the efficiency of computation is crucial. By formalizing general reasoning problems as DAG evaluation, we show that while CoT can solve these problems using a small number of parameters, *Latent Thought* (looped models) can solve them more efficiently (Sec. 3.2). To rigorously establish the separation, we extend our analysis to the computational complexity-theoretic setting (Sec. 3.3). We show that, given sufficient model size, looped models with $\log^k n$ loops achieve exactly the power of $\mathsf{TC}^k$ (Thm. 3.12), a canonical parallel computation class. By contrast, CoT with $\log^k n$ steps falls short of $\mathsf{TC}^k$ (Thm. 3.13), leading to a strict separation in computational power favoring Latent Thought (Thm. 3.16). In summary, Latent Thought gains efficiency by exploiting parallelism, whereas CoT remains fundamentally sequential.

**Chain of Thought Enhances Distribution Modeling**  The primary goal of language modeling is to represent a probability distribution over strings. We show that the stochastic decoding process in CoT fundamentally enhances its ability to model such distributions, surpassing what deterministic looped models can achieve (Thm. 4.9). Unlike prior work that focuses on exactly representing distributions, we instead analyze the ability of these models to *approximately* model structured distributions. This shift in problem formulation allows us to leverage classical results in computational complexity theory on approximate sampling (Jerrum et al., 1986) to derive a rigorous separation.

## 2 PRELIMINARIES AND RELATED WORK

### 2.1 MODELS OF COMPUTATION

We study a model in which a Transformer (Vaswani et al., 2017) is applied iteratively (formal definitions are given in Appendix B). Informally, the CoT model generates intermediate reasoning steps explicitly as tokens in an autoregressive manner before producing the final answer.

**Definition 2.1** (CoT). Let $\mathcal{V}$ be a vocabulary, and let $f_{\mathrm{dec}} : \mathcal{V}^* \to \mathcal{V}$ denote an *autoregressive*, decoder-only Transformer that generates the next token either deterministically or stochastically. Given an input sequence $x = (x_1, \ldots, x_n) \in \mathcal{V}^n$, define the intermediate outputs recursively by

$$f_{\mathrm{cot}}^0(x) := x, \quad f_{\mathrm{cot}}^{k+1}(x) := f_{\mathrm{cot}}^k(x) \cdot f_{\mathrm{dec}}(f_{\mathrm{cot}}^k(x)),$$

where $\cdot$ denotes concatenation. We define the output to be the last $L(n)$ tokens of $f_{\mathrm{cot}}^{T(n)}(x) \in \mathcal{V}^{n+T(n)}$, where $L(n)$ denotes the output length, i.e., $f_{\mathrm{cot}}^{T(n)}(x)_{n+T(n)-L(n)+1 : n+T(n)}$.

In contrast to CoT, looped TF iteratively refines internal embeddings without generating tokens.

**Definition 2.2** (looped Transformer). Let $f : \mathbb{F}^{d \times n} \to \mathbb{F}^{d \times n}$ denote a Transformer block, where $\mathbb{F}$ denotes finite-precision floating-point set and $d$ is the embedding dimension. Given an input sequence $x = (x_1, \ldots, x_n) \in \mathcal{V}^n$, define the sequence of hidden states recursively by

$$f_{\mathrm{loop}}^0(x) := (e(x_1), \ldots, e(x_n)), \quad f_{\mathrm{loop}}^{k+1}(x) := f(f_{\mathrm{loop}}^k(x)),$$

where $e : \mathcal{V} \to \mathbb{F}^d$ denotes an embedding function. After $T(n)$ loop iterations, the output is defined as the last $L(n) \le n$ tokens $\mathrm{Dec}(f_{\mathrm{loop}}^{T(n)}(x))_{n-L(n)+1 : n}$, where Dec denotes the decoding function.

### 2.2 RELATED WORK (SUMMARIZED IN TABLE 1)

**Computational Power of Chain-of-Thought**  Feng et al. (2023) first showed that CoT makes it possible to solve problems that would otherwise be intractable by characterizing its expressivity within a complexity-theoretic framework. Recent studies have further examined how its expressivity

Table 1: Summary of prior theoretical analyses on the computational power of CoT and looped TF.

| Paper | Model | Type | Problem Setting |
|---|---|---|---|
| Feng et al. (2023) | CoT | Det. | Mathematics & Decision-making |
| Merrill & Sabharwal (2024) | CoT | Det. | Automata & Turing machine |
| Li et al. (2024) | CoT | Det. | Boolean circuit |
| Nowak et al. (2024) | CoT | Pro. | Language modeling (PTM) |
| Saunshi et al. (2025) | looped | Det. | Non-looped, CoT, and Automata |
| Merrill & Sabharwal (2025a) | looped | Det. | Automata & Graph connectivity |
| Merrill & Sabharwal (2025b) | looped | Det. | Uniform $\mathsf{TC}^k$ |
| **Ours** | looped &CoT | Det. Pro. | DAGs & Non-uniform $\mathsf{TC}^k$ FPRAS & FPAUS |

scales with the number of reasoning steps (Merrill & Sabharwal, 2024; Li et al., 2024), comparing it with well-studied complexity classes such as automata, Turing machines, and circuit classes. While these studies simplify CoT decoding to a deterministic process and therefore focus only on deterministic decision problems, Nowak et al. (2024) analyzed probabilistic CoT and characterized its expressive power in representing probability distributions over strings.

**Computational Power of looped Transformer**   looped TFs are Turing-complete (Giannou et al., 2023), capable of simulating iterative algorithms (Yang et al., 2024) and graph algorithms (de Luca & Fountoulakis, 2024), whereas recent work (Liang et al., 2024) indicates that looped ReLU networks already yield strong expressive capacity. Sanford et al. (2024b) showed that a logarithmic-depth transformer, effectively a looped architecture, can simulate parallel computation. In comparison with CoT, Saunshi et al. (2025) showed that looped TF can express any computation performed by *deterministic* CoT, establishing the inclusion. Furthermore, Merrill & Sabharwal (2025a) proved that looped TF with $\log n$ iterations can solve problems that uniform CoT with $\log n$ steps cannot, thereby demonstrating a strict separation in logarithmic time. Concurrent work (Merrill & Sabharwal, 2025b) shows that fixed looped TF, with padding and polylogarithmic loops, is equivalent to uniform $\mathsf{TC}^k$. In contrast, we allow the model size to scale with the input size for asymptotic analysis.

## 3   LATENT THOUGHT ENABLES EFFICIENT PARALLEL SOLUTIONS

This section analyzes the separation between CoT and looped TF by formalizing deterministic computations as graph evaluation problems. In Section 3.2, we illustrate how the two models behave differently when evaluating the same computation, providing intuitive insight into their contrasting capabilities. Building on these observations, Section 3.3 formalizes the results within circuit complexity theory, characterizing their expressive power and establishing their separation.

### 3.1   PROBLEM SETTING

We formalize deterministic computations as directed acyclic graphs (DAGs), since straight-line programs can be represented in this form (Aho & Ullman, 1972). Example in Fig. 1(a).

**Definition 3.1** (Computation graph). Let $\Sigma$ be a finite alphabet, and let $\mathcal{F}$ denote a finite set of functions $f : \Sigma^* \to \Sigma$. A *computation graph* is a directed acyclic graph $G_n = (V_n, E_n)$ that defines a function $F_{G_n} : \Sigma^n \to \Sigma^{m(n)}$, where $m(n)$ denotes the output length. Here $V_n$ denotes the set of nodes, consisting of (i) $n$ input nodes with in-degree 0, (ii) function nodes labeled by $f \in \mathcal{F}$, which take as arguments the predecessor nodes specified by their incoming edges in $E_n$, and (iii) $m(n)$ output nodes with out-degree 0. The overall function is obtained by evaluating the graph in topological order. The *size* of the graph is $|V_n|$, denoted by $\mathrm{size}(G_n)$, and its *depth* is the length of the longest path from an input to an output node, denoted by $\mathrm{depth}(G_n)$

**Assumptions on models.**   Our goal is to evaluate the computational efficiency of each model through an asymptotic analysis of how the number of loops or steps scales with the input size $n$. In addition to this *time complexity*, we also consider the *space complexity*, such as the embedding

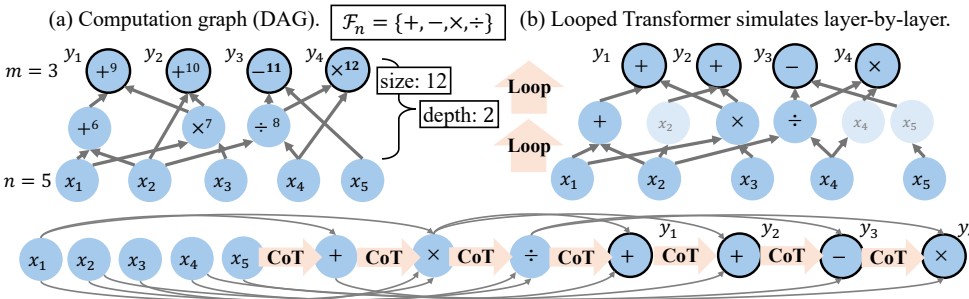

(c) Chain-of-Thought can simulate node by node.

Figure 1: Comparison of strategies for evaluating DAG. (a) A computation graph $G_n$. (b) A looped TF simulates the graph in a number of loops proportional to the depth of the graph $O(\mathrm{depth}(G_n))$. (c) A CoT simulates in a number of steps proportional to the size of the graph $O(\mathrm{size}(G_n))$.

size in Transformers, which also scales with the input size, analogous to the number of processors in classical parallel computation models. To this end, we adopt a *non-uniform* model, allowing a different model for each input size. This setting is consistent with prior work in the context of circuit complexity and parallel computation models (Sanford et al., 2024b; Li et al., 2024).

### 3.2 CoT Suffices with Size-scaled Steps and looped TF Suffices with Depth-scaled Loops

We show how each model can evaluate DAGs, which provides a lower bound on their expressivity. Before presenting our main result, we first state the underlying assumptions.

**Definition 3.2** (Merrill & Sabharwal, 2023). Informally, a model is *log-precision* if each scalar is stored with $O(\log n)$ bits and every arithmetic operation is rounded to that precision.

**Assumption 3.3** (Polynomial-size graph). $\mathrm{size}(G_n) \in \mathrm{poly}(n)$.

**Assumption 3.4** (Polynomially-efficient approximation, cf. (Feng et al., 2023)). Informally, each node function of $G_n$ can be approximated by a log-precision feedforward network whose parameter size is polynomial in the input length and the approximation error. We denote by $\mathrm{ff\_param}(G_n)$ an upper bound such that every $f \in \mathcal{F}$ admits such a network with at most $\mathrm{ff\_param}(G_n)$ parameters.

We first consider CoT. Under Assumptions 3.3, 3.4, a CoT model can simulate the computation by sequentially decoding node by node. Intermediate tokens act as scratchpad memory to store partial results, allowing the model to evaluate any node once all its predecessors are available.

**Theorem 3.5** (CoT for DAGs). *Let $\{G_n\}_{n\in\mathbb{N}}$ be a family of computation graphs that satisfy Assumptions 3.3 and 3.4. Then, for each $n \in \mathbb{N}$, there exists a log-precision CoT with parameter size bounded by $O(\mathrm{ff\_param}(G_n))$, such that for every input $x \in \Sigma^n$, the model outputs $F_{G_n}(x)$ in a number of steps proportional to the "size" of the graph, $\mathrm{size}(G_n)$.*

*Proof sketch.* The construction generalizes mechanisms in (Feng et al., 2023; Li et al., 2024): a CoT model can evaluate a graph by processing its nodes in topological order. At each step, the attention mechanism retrieves the outputs of the predecessor nodes, which are available as previously generated tokens. A feed-forward network is then applied to compute the output of the current node, and this value is generated as the next token. The iterative process yields the final result in $O(|V_n|)$ steps. □

Unlike CoT, looped TF can simulate the computation in parallel, layer by layer, where all nodes at the same depth are computed simultaneously, provided the embedding size is sufficiently large.

**Assumption 3.6** (Output bound). The computation graph contains at most $n$ output nodes.

**Theorem 3.7** (looped TF for DAGs). *Let $\{G_n\}_{n\in\mathbb{N}}$ be a family of computation graphs that satisfy Assumptions 3.3, 3.4, and 3.6. Then, for each $n \in \mathbb{N}$, there exists a log-precision looped TF with parameter size $O(\mathrm{ff\_param}(G_n) \cdot \mathrm{size}(G_n))$, such that for every input $x \in \Sigma^n$, it computes $F_{G_n}(x)$ in a number of loops proportional to the "depth" of the graph $G_n$, , $\mathrm{depth}(G_n)$.*

*Proof sketch.* The role assignment of each layer is based on Li et al. (2024), as illustrated in Figure 5. A uniform attention layer aggregates all inputs from the entire sequence into a single position. The hidden state is partitioned into dedicated blocks, each block corresponding to a node in the computation graph, and the looped feedforward layer simultaneously computes the local function of every node on its respective block. By iterating this process, all nodes at the same depth of the graph are evaluated in parallel, and successive iterations propagate the computation layer by layer. □

**Remark.**  Formal proofs are given in Appendix C, with illustrations in Fig. 1. These results show distinct strengths: CoT can employ intermediate steps as scratchpad memory, supporting general computation without constraints. In contrast, looped TF can exploit structural parallelism to achieve greater efficiency, but requires additional resources.

### 3.3 Formal Characterization of Expressive Power and Separation

To establish the separation, we formalize the models in terms of their corresponding complexity-theoretic classes. We begin by defining the model classes as in Li et al. (2024).

**Definition 3.8** (CoT and Loop). Let $\mathsf{CoT}[T(n), d(n), s(n)]$ (resp. $\mathsf{Loop}[T(n), d(n), s(n)]$) denote the set of languages $\mathcal{L} : \{0, 1\}^* \to \{0, 1\}$ for which there exists a *deterministic* CoT (resp. looped TF) $M_n$ for each input size $n$, with embedding size $O(d(n))$ and $O(s(n))$ bits of precision, such that for all $x \in \{0, 1\}^n$, the final output token, after $O(T(n))$ decode steps (resp. loops), is $\mathcal{L}(x)$.

We adopt standard circuit complexity classes as the point of comparison.

**Definition 3.9** (Circuit Classes).  All circuit families are assumed to be non-uniform. Unless otherwise stated, circuits use bounded-fanin Boolean gates (AND, OR, or NOT).

- $\mathsf{SIZE}[s(n)]$: Languages decidable by circuit families of size at most $O(s(n))$.

- $\mathsf{DEPTH}[d(n)]$: Languages decidable by circuit families of depth at most $O(d(n))$.

- $\mathsf{NC}^k$: Languages decidable by circuit families of size $\mathsf{poly}(n)$ and depth $O(\log^k n)$.

- $\mathsf{AC}^k$: Same as $\mathsf{NC}^k$, except that unbounded-fanin AND/OR gates are allowed.

- $\mathsf{TC}^k$: Same as $\mathsf{AC}^k$, except that threshold gates of unbounded fanin are allowed.

Specifically, $\mathsf{NC}^k$, $\mathsf{AC}^k$, and $\mathsf{TC}^k$ characterize problems that admit efficient parallel algorithms running in polylogarithmic time using polynomially many processors (Stockmeyer & Vishkin, 1984). First, the results for DAGs presented in the previous section can be formalized.

**Theorem 3.10** (Li et al., 2024). $\forall T(n) \in \mathsf{poly}(n)$, $\mathsf{SIZE}[T(n)] \subseteq \mathsf{CoT}[T(n), \log n, 1]$.

**Theorem 3.11.** $\forall T(n) \in \mathsf{poly}(n)$, $\mathsf{DEPTH}[T(n)] \subseteq \mathsf{Loop}[T(n), \mathsf{poly}(n), 1]$.

Based on Theorem 3.11, when given sufficiently large model size, looped Transformers not only achieve but exactly match the computational power of the parallel class $\mathsf{TC}^k$.

**Theorem 3.12.** $\forall k \in \mathbb{N}$, $\mathsf{Loop}[\log^k n, \mathsf{poly}(n), 1 \text{ (resp. } \log n)] = \mathsf{AC}^k \text{ (resp. } \mathsf{TC}^k)$.

*Proof sketch.* The inclusion from circuits to looped TFs follows from Theorem 3.7. For the converse inclusion, we build on the arguments of Li et al. (2024), which showed that under finite precision, $\mathsf{CoT}[0, \mathsf{poly}(n), 1 \text{ (resp. } \log n)] \subseteq \mathsf{AC}^k \text{ (resp. } \mathsf{TC}^k)$. We extended this to the looped setting, yielding the desired inclusion. The formal proof is provided in Appendix C.6. □

Conversely, we establish an upper bound on CoT: while CoT can simulate computations gate-by-gate, its inherently sequential structure fundamentally limits efficient modeling of parallel computation.

**Lemma 3.13.** $\forall k \in \mathbb{N}$, $\mathsf{CoT}[\log^k n, \mathsf{poly}(n), \log n] \subseteq \mathsf{TC}^{k-1}$.

*Proof.* The total $\log^k n$ steps can be divided into $\log^{k-1} n$ blocks, each consisting of $\log n$ steps. Since $\mathsf{CoT}[\log n, \mathsf{poly}(n), \log n] \subseteq \mathsf{TC}^0$ (Li et al., 2024), each block with the previous block's outputs fed as inputs to the next block can be simulated in $\mathsf{TC}^0$; iterating this over $\log^{k-1} n$ layers yields a depth $O(\log^{k-1} n)$ threshold circuit, i.e., in $\mathsf{TC}^{k-1}$. □

This leads to the conclusion that CoT is subsumed by looped transformers:

**Corollary 3.14.** $\mathsf{CoT}[\log^k n, \mathrm{poly}(n), \log n] \subseteq \mathsf{Loop}[\log^{k-1} n, \mathrm{poly}(n), 1]$.

Moreover, under standard complexity-theoretic assumptions, we obtain a separation.

**Theorem 3.15.** $\mathsf{TC}^{k-1} \subsetneq \mathsf{NC}^k \Rightarrow \mathsf{CoT}[\log^k n, \mathrm{poly}(n), \log n] \subsetneq \mathsf{Loop}[\log^k n, \mathrm{poly}(n), 1]$.

**Theorem 3.16.** $\mathsf{TC}^{k-1} \subsetneq \mathsf{TC}^k \Rightarrow \mathsf{CoT}[\log^k n, \mathrm{poly}(n), \log n] \subsetneq \mathsf{Loop}[\log^k n, \mathrm{poly}(n), \log n]$.

**Remark.** The claims follow directly from Theorem 3.12 and Lemma 3.13. These results show that, under the same polylogarithmic number of steps or loops, looped TF can simulate parallel computations more efficiently than CoT. This highlights the inherent parallelism of looped models.

## 4 CHAIN-OF-THOUGHT ENABLES APPROXIMATE SAMPLING

This section presents the converse result, showing a setting in which CoT has a clear advantage. We study the task of approximately sampling structured distributions over strings (Section 4.1). Section 4.2 introduces the computational complexity background for approximate sampling. Building on this foundation, Section 4.3 provides a formal separation demonstrating that CoT can leverage stochastic decoding to surpass deterministic looped computation in distribution modeling.

### 4.1 PROBLEM SETTING

We study the task of approximately sampling from a target distribution over strings, a standard lens for analyzing the expressive power of language-modeling architectures (Lin et al., 2021; Nowak et al., 2024). For an input $x \in \Sigma^*$, let $p(\cdot \mid x)$ denote the target distribution and and $q(\cdot \mid x)$ the distribution induced by a given model. Given $x$, the model generates an output sequence $(y_1, \ldots, y_m) \in \Sigma^m$.

**Definition 4.1** (Approximate Sampling). We say that $q(\cdot \mid x)$ is an $\varepsilon$-approximation of $p(\cdot \mid x)$ if $\left\| q(\cdot \mid x) - p(\cdot \mid x) \right\|_{\mathrm{TV}} \leq \varepsilon$, where $\| \cdot \|_{\mathrm{TV}}$ denotes total variation distance.

We consider two computational models for sequence generation via next-token prediction for CoT and looped transformers. Specifically, CoT stochastically generates intermediate reasoning steps before producing the target symbol $y_i$. In contrast, looped TF performs additional iterations in the latent space and samples the target symbol $y_i$ after the final loop iteration, as in practice (Csordás et al., 2024; Snell et al., 2025; Bae et al., 2025). Definitions are in Appendix D.1.

### 4.2 PRELIMINARIES ON COMPLEXITY OF APPROXIMATE SAMPLING

The complexity of approximate sampling has been extensively studied in the context of *uniform generation of combinatorial structures*. Let $R \subseteq \Sigma^* \times \Sigma^*$ be a relation. For an input $x \in \Sigma^*$, define the solution set $R(x) = \{ y \in \Sigma^* : (x, y) \in R \}$.

**Definition 4.2** (FPAUS). Uniform generation asks to sample $y$ uniformly from $R(x)$. A *fully polynomial almost uniform sampler (FPAUS)* is a randomized algorithm that, given $x \in \Sigma^*$ and $\varepsilon > 0$, runs in time polynomial in $|x|$ and $\log(1/\varepsilon)$, and outputs a distribution $q(\cdot \mid x)$ such that $\left\| q(\cdot \mid x) - U(R(x)) \right\|_{\mathrm{TV}} \leq \varepsilon$, where $U(R(x))$ denote the uniform distribution over $R(x)$.

The complexity of approximate sampling is closely related to the problem of approximating the size of the solution space $f(x) = |R(x)|$, i.e., approximate counting:

**Definition 4.3** (FPTAS). An algorithm is a *fully polynomial-time approximation scheme (FPTAS)* for $f : \Sigma^* \to \mathbb{N}$ if, for any input $x$ and $\varepsilon > 0$, it outputs $\hat{f}(x)$ such that $(1 - \varepsilon)f(x) \leq \hat{f}(x) \leq (1 + \varepsilon)f(x)$, and runs in time polynomial in $|x|$ and $1/\varepsilon$.

In some cases, however, deterministic approximation remains intractable. By contrast, randomization enables efficient approximation by allowing a small probability of error (Karp & Luby, 1983):

**Definition 4.4** (FPRAS). An algorithm is a *fully polynomial-time randomized approximation scheme (FPRAS)* for $f$ if, given $\varepsilon > 0$ and $\delta > 0$, it outputs $\hat{f}(x)$ satisfying

$$\Pr\left[ (1 - \varepsilon)f(x) \leq \hat{f}(x) \leq (1 + \varepsilon)f(x) \right] \geq 1 - \delta,$$

and runs in time polynomial in $|x|$, $1/\varepsilon$, and $\log(1/\delta)$.

For example, the knapsack problem admits an FPTAS using a dynamic programming algorithm (Vazirani, 2001), and the DNF counting is only known to admit an FPRAS (Karp et al., 1989). Complexity theory shows that approximate counting and approximate sampling are computationally equivalent:

**Theorem 4.5** (Jerrum et al., 1986). *If $R$ is self-reducible, then there exists an FPRAS for approximating $|R(x)|$ if and only if there existsstates for sampling uniformly from $R(x)$.*

Here the *self-reducibility* refers to the property that solutions can be reduced to smaller subproblems:

**Definition 4.6** (Informal: Self-reducibility (Schnorr, 1976)). A relation $R$ is *self-reducible* if there exists a polynomial-time procedure that, given any input $x$ and prefix $y_{1:k}$ (with respect to a fixed output order), produces a sub-instance $\psi(x, y_{1:k})$ such that every solution $z$ of $\psi(x, y_{1:k})$ extends $y_{1:k}$ to a solution of $R(x)$ (and conversely), i.e., $\mathrm{solution}\big(\psi(x, y_{1:k})\big) = \{ z \mid \mathrm{concat}(y_{1:k}, z) \in R(x) \}$.

### 4.3 Separation in Approximate Sampling Capability

We first analyze the expressiveness of CoT. While algorithms such as FPAUS (and FPRAS) regard the accuracy parameter $\varepsilon$ as part of the input, this contrasts with the behavior of practical LLMs: once a base model has been pretrained to sufficiently low error, it can meet the required accuracy $\varepsilon$ at inference time. To formalize this, we first formally define well-pretrained models.

**Definition 4.7** (Pretraining convergence). Let $p$ be a target distribution supported on $\Sigma^*$, let $\alpha(n) \in \mathrm{poly}(n)$, and let $\gamma \in (0, \frac{1}{2})$. A stochastic model $\pi_\omega$, indexed by the to be an $(\alpha, \gamma)$-*weak probable approximation* to $p$ if, for every position $i \in \{1, \ldots, m\}$, every input $x \in \Sigma^*$, and every prefix $y_{<i} := (y_1, \ldots, y_{i-1})$, it holds that

$$\Pr\left[(1 - \tfrac{1}{\alpha(|x|)})p(y_i \mid x, y_{<i}) \;\leq\; \pi_\omega(y_i \mid x, y_{<i}) \;\leq\; (1 + \tfrac{1}{\alpha(|x|)})p(y_i \mid x, y_{<i})\right] \;\geq\; \tfrac{1}{2} + \gamma.$$

Intuitively, the model does not guarantee a small approximation error in every run, but achieves with probability strictly greater than random chance.

**Proposition 4.8.** *Let $R$ be a relation. Suppose the CoT model $\pi$ is an $(\alpha, \gamma)$-weak probable approximation to the uniform distribution $U\big(R(x)\big)$ for a sufficiently large $\alpha$. Then there exists a polynomial-time algorithm, with access to $\pi$, that constitutes an FPAUS.*

We now establish a conditional separation between CoT and looped TFs. Building on the result of Nowak et al. (2024), we show that CoT can represent any FPRAS, and therefore can realize convergence during pretraining, and then constitute an FPAUS as above. In contrast, we show that there exist instances where a looped TF fails to achieve comparable approximation guarantees.

**Theorem 4.9.** *Assume* FPTAS $\subsetneq$ FPRAS *for self-reducible relations $R$, with respect to the counting problem $|R(x)|$. Then there exists a function $\alpha(n) \in \mathrm{poly}(n)$ and constant $\gamma > 0$ such that a CoT model with polynomial steps satisfies the $(\alpha, \gamma)$-weak probable approximation to the uniform distribution $U\big(R(x)\big)$, whereas no looped TF with polynomial loops can achieve the same approximation, nor realize an FPAUS for $U\big(R(x)\big)$.*

*Proof sketch.* Suppose, toward a contradiction, that for every polynomial function, there exists a looped TF satisfying the weak probable approximation. Since a looped TF introduces randomness only at the decoding stage, its logits are computed deterministically, and the stochasticity can thus be derandomized. This would yield an FPTAS, contradicting the assumption that FPTAS $\subsetneq$ FPRAS. □

**Remark.** Formal proofs are given in Appendix D. This result suggests that looped models may not achieve the same expressive power as CoT in approximation under the assumption, uncovering an advantage of CoT and the benefit of stochastic decoding for enabling randomized approximation.

## 5 Experiments

This section provides empirical validation of our theoretical results. First, we evaluate parallelizable reasoning tasks, where our results show that looped TFs can solve these problems with substantially

fewer iterative operations than CoT, consistent with the separation established in Section 3. Second, we consider a computationally hard approximate counting task, where the advantage shifts: CoT's stochastic sampling enables accurate approximation, whereas deterministic looped TFs fail to match its performance, supporting the theoretical results in Section 4.

## 5.1 EXPERIMENTAL SETTING

This section provides the task descriptions and training methodology for both tasks.

### 5.1.1 PARALLELIZABLE TASKS

**Task Description**   We use four problems. (1) Word problems for finite non-solvable groups: given a sequence of generators, the task is to evaluate their composition. This problem is $\mathsf{NC}^1$-complete (Barrington, 1986), also studied for looped TF by Merrill & Sabharwal (2025a). (2) $s$–$t$ connectivity (STCON): given a directed graph $G = (V, E)$ and two vertices $s, t \in V$, the task is to decide whether $t$ is reachable from $s$. This problem belongs to $\mathsf{TC}^1$ (Gibbons & Rytter, 1989). (3) Arithmetic expression evaluation: given a formula consisting of $+, \times, -, /$ operations on integers, the task is to evaluate it. This problem is $\mathsf{TC}^0$-reducible to Boolean formula evaluation (Feng et al., 2023), which is $\mathsf{NC}^1$-complete (Buss, 1987). (4) Edit distance: given two strings $x$ and $y$, the task is to compute the minimum cost to transform $x$ into $y$. By reducing the dynamic programming formulation to shortest paths, this problem is in $\mathsf{TC}^1$ (Apostolico et al., 1990). Details are in Appendix E.1.1.

**Training Methodology**   For both CoT and looped, the same Transformer block is adopted to ensure comparable parameter counts. For CoT models, training is performed with supervision of sequential algorithms. For instance, in dynamic programming tasks, the model is supervised to emit DP table entries. For inference, only the input is given. To control the number of CoT steps, intermediate positions are uniformly sampled, while preserving structure, along the full solution trajectory. All algorithms for each task are provided in Appendix E.1.2. We found that, just as CoT benefits from intermediate supervision, looped similarly benefits from curriculum learning for certain tasks, as also discussed by Merrill & Sabharwal (2025a). Details of configuration are in Appendix E.1.3.

### 5.1.2 APPROXIMATION TASK

**Task Description**   We evaluate approximate reasoning using a DNF counting task, exploiting the fact that approximate counting and approximate sampling are equivalent for self-reducible relations. Given a DNF formula $F$ with $m$ clauses over $n$ Boolean variables, the goal is to estimate the number of satisfying assignments $|F|$. Although exact DNF counting is #P-complete, an FPRAS is known.

**Training Methodology**   For looped TF, we trained the models end-to-end to directly map inputs to counts. In contrast, CoT models were trained with intermediate steps simulating a single trial of the randomized counting algorithm of Karp & Luby (1983). Performance is evaluated using relative error. At inference time, CoT executes multiple Monte Carlo samples and aggregates the results through post-processing. The total number of CoT steps is defined as the product of the reasoning steps per sample and the number of Monte Carlo samples. Further details are provided in Appendix E.2.

## 5.2 RESULTS

Table 2 presents results on parallelizable tasks, comparing looped TF and CoT under varying loop or step counts. Across all tasks, looped TF solves the problems with only a small number of loops, whereas CoT requires substantially more steps to reach the same performance. These results support our theoretical results on the separation between looped TF and CoT for parallel computation.

We also follow the evaluation protocol of measuring performance across varying input sizes and loop counts, a methodology also adopted in prior studies on word problems (Merrill & Sabharwal, 2025a) and $k$-hop tasks (Sanford et al., 2024b). Figure 2 presents our results for looped TFs, illustrating that as the input size $n$ increases, the number of loops required to maintain high accuracy grows only logarithmically. This empirical trend aligns with our theoretical claim that a (poly-)logarithmic loop count suffices, thereby demonstrating that looped TFs can efficiently simulate parallel computation.

Table 2: Accuracy on parallelizable tasks, with varying loop/step counts. looped TF solves the tasks with fewer loops, whereas CoT requires more steps. Here, $n$ denotes the input size. $n = 32/16$ indicates that looped TF uses $n = 32$ and CoT uses $n = 16$.

| Task | $n$ | looped TF (loops) | | | | CoT (steps) | | | |
|---|---|---|---|---|---|---|---|---|---|
| | | 2 | 4 | 6 | 8 | 8 | 16 | 32 | 64 |
| Word Problem | 64 | 0.8 | 0.8 | **100.0** | **100.0** | 0.8 | 0.8 | 0.8 | **100.0** |
| Graph Connectivity | 32 | 80.8 | 95.8 | **99.0** | **99.0** | 81.0 | 81.4 | 83.6 | **100.0** |
| Arithmetic Evaluation | 32/16 | 43.7 | 99.4 | 99.5 | **99.7** | 41.0 | 41.5 | 41.2 | **82.5** |
| Edit Distance | 32/16 | 57.3 | 72.9 | 86.2 | **90.7** | 69.2 | 70.3 | 82.6 | **94.8** |

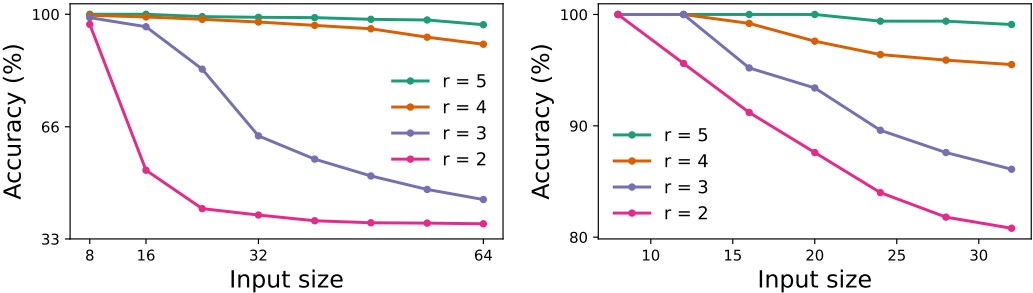

Figure 2: Accuracy of $r$-looped TFs on the arithmetic evaluation (left) and the connectivity (right). Each curve shows the performance for a fixed loop count $r$ as the input size $n$ increases. Increasing the loop count improves accuracy, and a logarithmic number of loops suffices.

These results extend prior work in several directions. For the word problem, Merrill & Sabharwal (2025a) examined looped and standard Transformers, but our results show that a small number of CoT steps is insufficient to solve it. For graph tasks of Sanford et al. (2024b), we evaluate larger input sizes and show that looped models remain sufficient. For arithmetic evaluation and edit distance, while Feng et al. (2023) demonstrated that CoT suffices with quadratic steps, our results show that looped TF can solve them more efficiently.

Figure 3 shows the result on the approximation task. The CoT model achieves approximate inference via Monte Carlo sampling, and its relative error decreases markedly when provided with sufficient reasoning steps and samples. In contrast, the looped TF fails to learn even as the number of loops increases, with the error remaining constant because the model outputs identical predictions across all inputs. These findings support our theoretical claim on the superiority of CoT in randomized approximation.

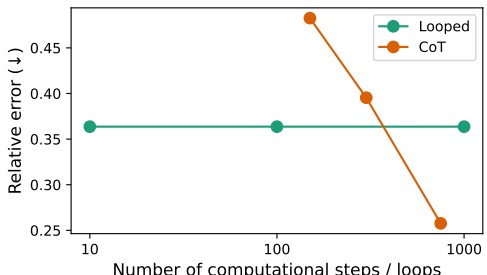

Figure 3: Relative error on counting task with respect to the number of steps or loops.

## 6 CONCLUSION

We rigorously characterize the respective computational strengths of chain-of-thought and latent thought from a complexity-theoretic standpoint. Our analysis delineates the precise capabilities and fundamental limitations of looped language models, thereby establishing a principled theoretical foundation for their continued development. While our results are supported by experiments on synthetic tasks whose complexity is precisely characterized, future work will investigate whether the same holds on realistic benchmarks. Future work will explore to what extent these strengths can be simultaneously realized in alternative architectures, such as diffusion language models.

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

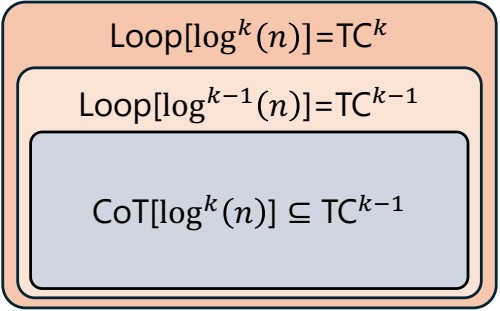
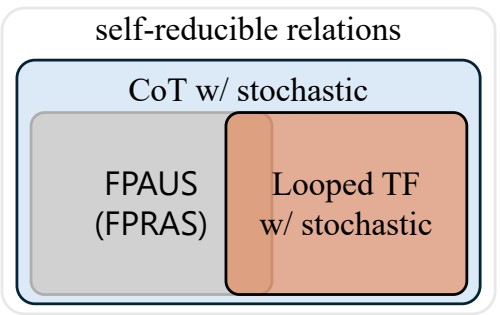

Figure 4: Overview of the results. Left: deterministic setting, illustrating the relationship between looped TFs with $\log^{k-1}(n)$ and $\log^k(n)$ loops, CoT with $\log^k(n)$ steps, and the corresponding circuit classes ($\mathsf{TC}^k$, $\mathsf{TC}^{k-1}$), all under the assumptions of $\log(n)$-precision and $\mathrm{poly}(n)$ size. Right: stochastic setting for a self-reducible relation $R$, where CoT with stochastic decoding realizes FPAUS for the uniform distribution over $R(x)$ or an FPRAS for $|R(x)|$ within $\mathrm{poly}(n)$ steps, whereas looped TFs with stochastic decoding fail to achieve the same within $\mathrm{poly}(n)$ loops.

## A   NOTATION

Vectors are written in lowercase bold letters (e.g., $\boldsymbol{x}$) and matrices in uppercase bold letters (e.g., $\boldsymbol{W}$). The $i$-th entry of a vector $\boldsymbol{x}$ is $\boldsymbol{x}_i$, the vector from the $i$-th to the $j$-th entry is denoted by $\boldsymbol{x}_{i:j}$, and the $(i,j)$-th entry of a matrix $\boldsymbol{W}$ is $\boldsymbol{W}_{i,j}$. We use the symbol $*$ to denote a "don't care" value (or block of values). For $n \in \mathbb{N}^+$, let $[n] := \{1, 2, \ldots, n\}$. We sometimes write column vectors horizontally, e.g., $\boldsymbol{x} = (x_1, \ldots, x_n)$, for brevity. The Hadamard (element-wise) product is $\odot$. $\boldsymbol{e}_i \in \{0,1\}^d$ is the $i$-th standard basis vector, $\mathbf{1}_d \in \mathbb{R}^d$ (or $1_d$) the all-ones vector, and $\mathbf{0}_d \in \mathbb{R}^d$ the zero vector. $\boldsymbol{I}_d \in \mathbb{R}^{d \times d}$ denotes the $d \times d$ identity matrix, and $\mathbf{0}_{m \times n} \in \mathbb{R}^{m \times n}$ the $m \times n$ zero matrix. The indicator function is $\mathbf{1}[\cdot]$, and $\bigoplus$ denotes block-diagonal concatenation. Functions on scalars or vectors are written in upright letters (e.g., FFN), while functions on matrices are boldface (e.g., $\mathbf{ATTN}$). Boldface is also used when scalar- or vector-level functions are extended to sequence level and applied independently to each token (e.g., $\mathbf{FFN}$). Finally, $\mathrm{poly}(n)$ denotes the set of functions growing at most polynomially: $\mathrm{poly}(n) := \left\{ f : \mathbb{N} \to \mathbb{N} \mid \exists k \in \mathbb{N}, \ \exists c > 0, \ \forall n \in \mathbb{N}, \ f(n) \leq c \cdot n^k \right\}$.

## B   FORMAL DEFINITIONS FOR SECTION 2

### B.1   TRANSFORMER BLOCK

We define the computational components of a Transformer using the notation of Merrill & Sabharwal (2025a). Let $\mathbb{F}_s$ denote the set of $s$-bit fixed-point numbers with truncated arithmetic.

**Definition B.1** (Transformer). A Transformer consists of the following components:

1. A word embedding function WE : $\mathcal{V} \to \mathbb{F}_s^m$, where $\mathcal{V}$ denotes the vocabulary set.

2. A positional embedding function $\text{PE} : \mathbb{N} \to \mathbb{F}_s^m$.
3. A saturated multi-head self-attention layer $\mathbf{SA} : \mathbb{F}_s^{m \times N} \to \mathbb{F}_s^{m \times N}$ for arbitrary sequence length $N$, parameterized by a matrix $\boldsymbol{O} : \mathbb{F}_s^{s \times H} \to \mathbb{F}_s^m$ and, for each head $h \in [H]$ with head size $s$, matrices $\boldsymbol{Q}_h, \boldsymbol{K}_h, \boldsymbol{V}_h : \mathbb{F}_s^m \to \mathbb{F}_s^s$. Given an input $\boldsymbol{x}_i \in \mathbb{F}_s^m$ for each position $i \in [N]$, it computes the query $\mathbf{q}_{i,h} = \boldsymbol{Q}_h \boldsymbol{x}_i$, key $\boldsymbol{k}_{i,h} = \boldsymbol{K}_h \boldsymbol{x}_i$, and value $\mathbf{v}_{i,h} = \boldsymbol{V}_h \boldsymbol{x}_i$, and outputs $\boldsymbol{O} \cdot (\boldsymbol{a}_{i,1}, \ldots, \boldsymbol{a}_{i,H})$, where each attention output $\boldsymbol{a}_{i,h}$ is defined as:

$$\boldsymbol{a}_{i,h} = \sum_{j=1}^{c(i)} \frac{\exp(\mathbf{q}_{i,h}^\top \boldsymbol{k}_{j,h})}{Z_{i,h}} \cdot \mathbf{v}_{j,h}, \quad Z_{i,h} = \sum_{j=1}^{c(i)} \exp(\mathbf{q}_{i,h}^\top \boldsymbol{k}_{j,h}),$$

with $c(i) = i$ for causal attention and $c(i) = n$ for full attention.
4. A feedforward layer $\text{FF} : \mathbb{F}_s^m \to \mathbb{F}_s^m$ with parameter $\boldsymbol{W}_1 : \mathbb{F}_s^m \to \mathbb{F}_s^w$, $\boldsymbol{W}_2 : \mathbb{F}_s^w \to \mathbb{F}_s^m$, and $\boldsymbol{b} \in \mathbb{F}_s^m$, where $w$ is the hidden dimension. Given an input $\boldsymbol{x}_i \in \mathbb{F}_s^m$, it outputs $\boldsymbol{W}_2 \text{ReLU}(\boldsymbol{W}_1 \boldsymbol{x}_i + \boldsymbol{b})$, where $\text{ReLU}(\boldsymbol{x}) = (\max\{0, \boldsymbol{x}_1\}, \ldots, \max\{0, \boldsymbol{x}_m\})^\top$.
5. An output function $\mathbf{OUT} : \mathbb{F}_s^m \to \mathbb{F}_s^{|\mathcal{V}|}$, parameterized as a linear transformation.

## B.2 CHAIN-OF-THOUGHT

**Definition B.2** (CoT). Let the Transformer be defined as the composition:

$$f_{\text{dec}} \coloneqq \mathbf{OUT} \circ (\text{id} + \mathbf{FF}_L) \circ (\text{id} + \mathbf{SA}_L) \circ \cdots \circ (\text{id} + \mathbf{FF}_1) \circ (\text{id} + \mathbf{SA}_1) \circ (\mathbf{WE} + \mathbf{PE}),$$

where $\mathbf{SA}_\ell$ and $\mathbf{FF}_\ell$ denote the causal attention and the feedforward layers at depth $\ell \in [L]$, respectively, and id denotes the identity function. The input tokens are first embedded via the word embedding function $\mathbf{WE}$ and the positional encoding $\mathbf{PE}$, and the final output is produced by a linear projection $\mathbf{OUT}$. Given an input sequence $x = (x_1, \ldots, x_n) \in \mathcal{V}^n$, we define the initial sequence as: $f_{\text{cot}}^0(x) \coloneqq x$. Then, the *CoT* computes recursively as:

$$f_{\text{cot}}^{k+1}(x) \coloneqq f_{\text{cot}}^k(x) \cdot \text{Dec}(f_{\text{dec}}(f_{\text{cot}}^k(x))),$$

where $\cdot$ denotes concatenation of sequences, and $\text{Dec}(\cdot)$ is a decoding function that maps the output logits to a token in $\mathcal{V}$: in the *deterministic* model, $\text{Dec}(z) \coloneqq \arg\max_{i \in [|\mathcal{V}|]} z_i$; in the *stochastic* model, $\text{Dec}(z) \sim \text{Multinomial}(z_i / \sum_j z_j)$, assuming $z_i > 0$ for all $i$. The final output of the CoT model after $T(n)$ steps is defined as the last output length $m$ tokens of $f_{\text{cot}}^{T(n)}(x)$.

## B.3 LOOPED TRANSFORMER

**Definition B.3** (looped TF). Let the Transformer block be defined as the composition:

$$f \coloneqq (\text{id} + \mathbf{FF}_L) \circ (\text{id} + \mathbf{SA}_L) \circ \cdots \circ (\text{id} + \mathbf{FF}_1) \circ (\text{id} + \mathbf{SA}_1),$$

where $\mathbf{SA}_\ell$ and $\mathbf{FF}_\ell$ denote the (non-causal) self-attention and feedforward layers at depth $\ell \in [L]$.

Given an input token sequence $x = (x_1, \ldots, x_n) \in \mathcal{V}^n$, the initial hidden state is: $f_{\text{loop}}^0(x) \coloneqq \mathbf{WE}(x)$. At each loop iteration $k$, the hidden state is updated by:

$$f_{\text{loop}}^{k+1}(x) \coloneqq f\left(f_{\text{loop}}^k(x)\right).$$

The final outputs after $T(n)$ loop iterations are decoded as $\text{Dec} \circ \mathbf{OUT} \circ f_{\text{loop}}^{T(n)}(x)$, and the model's prediction is defined as the last output length $m \leq n$ tokens of this projected sequence.

## C DEFERRED PROOFS FOR SECTION 3

### C.1 PRECISION MODELING

Following the definition of finite precision in Li et al. (2024), we focus here, for simplicity, on signed floating-point numbers without exponents.

**Definition C.1** (Floating-point Representation, cf. (Li et al., 2024)). Consider floating-point numbers with a mantissa part of $s$ bits and a sign bit of 1, totaling $(s + 1)$ bits. We denote the set of such floating-point numbers by $\mathbb{F}_s$, and define $B_s \triangleq \max \mathbb{F}_s$.

**Definition C.2** (Correct Rounding, cf. (Li et al., 2024)). For any $x \in \mathbb{R}$ and any closed subset $\mathbb{F} \subset \mathbb{R}$ containing 0, we define the *correct rounding* $\mathrm{round}(x, \mathbb{F})$ as the number in $\mathbb{F}$ closest to $x$. In particular, rounding to a floating-point number with mantissa part $s$ bits is denoted by $[\cdot]_s$. Rounding applied to vectors is to be operated coordinate-wise.

They also define primitive arithmetic under finite precision by applying rounding after each basic operation. In particular, for multi-operand operations, rounding is applied after each binary operation. Finite-precision summation over more than two numbers is thus defined as follows.

**Definition C.3** (Summation with Iterative Rounding (Li et al., 2024)). For any $s, n \in \mathbb{N}^+$ and vector $\boldsymbol{x} \in \mathbb{R}^n$, define the *summation with iterative rounding to $s$-bit precision*

$$\mathrm{sum}_s : \bigcup_{n \in \mathbb{N}^+} (\mathbb{F}_s)^n \to \mathbb{F}_s, \tag{1}$$

where, for any $n \in \mathbb{N}^+$ and $\boldsymbol{x} = (x_1, \dots, x_n) \in \mathbb{R}^n$,

$$\mathrm{sum}_s(x) := \Big[ \Big[ \cdots \Big[ [x_1 + x_2]_s + x_3 \Big]_s + \cdots + x_{n-1} \Big]_s + x_n \Big]_s. \tag{2}$$

Based on this definition, all computations in the Transformer block of Definition B.1 can be represented in finite precision. The inner product and the matrix product are defined as

$$\boldsymbol{x}^\top \boldsymbol{y} := \mathrm{sum}_s(\boldsymbol{x} \odot \boldsymbol{y}), \qquad (\boldsymbol{AB})_{i,j} := \boldsymbol{A}_{i,:}^\top \boldsymbol{B}_{:,j}. \tag{3}$$

Throughout this section, we interpret all operations as finite-precision computations as defined above.

## C.2 Definition of Assumption 3.4

Our definition of polynomially-efficient approximation is based on that of Feng et al. (2023), but differs in domain. Whereas their focus is on real-valued functions, ours concerns symbolic functions.

**Definition C.4** (Polynomially-efficient approximation). We say that a function $f_n : \Sigma^{\ell(n)} \to \Sigma$ admits a *polynomially-efficient approximation* if there exists a log-precision feedforward network $\mathrm{FF} : \mathbb{F}_{s(n)}^{\ell(n) \cdot |\Sigma|} \to \mathbb{F}_{s(n)}^{|\Sigma|}$, $s(n) = O(\log n)$, such that the following holds: for every input $\boldsymbol{x} = (x_1, \dots, x_{\ell(n)}) \in \Sigma^{\ell(n)}$, every error tolerance $\delta > 0$,

$$\mathrm{FF}(\boldsymbol{e}(x_1), \dots, \boldsymbol{e}(x_{\ell(n)}))_i = \begin{cases} \geq 1 - \delta & \text{if } \boldsymbol{e}(f_n(\boldsymbol{x})) = \boldsymbol{e}_i, \\ \leq \delta & \text{else}, \end{cases} \tag{4}$$

where $\boldsymbol{e} : \Sigma \to \{0, 1\}^{|\Sigma|}$ denote the one-hot encoding. Moreover, the number of parameters of the feedforward network is bounded by a polynomial in $\ell(n)$ and $1/\delta$.

## C.3 Technical Lemmas

In this section, we provide the key components for our constructive proofs.

### C.3.1 Orthogonal Vectors

We follow the notation of Li et al. (2024). For any positive integer $s \in \mathbb{N}^+$ and $x \in \{0, 1, \dots, 2^s - 1\}$, we denote by $\mathrm{bin}_s(x) \in \{0, 1\}^s$ the standard binary representation of $x$ using $s$ bits, defined such that $x = \sum_{i=1}^s 2^i \cdot (\mathrm{bin}_s(x))_i$. We further define the signed binary encoding of $x$, denoted by $\mathrm{sbin}_s(x) \in \{-1, 1\}^s$, as $\mathrm{sbin}_s(x) = 2 \cdot \mathrm{bin}_s(x) - (1, \dots, 1)$. Let $\boldsymbol{x}, \boldsymbol{y} \in \mathbb{R}^s$ be two vectors of the same length. We define their interleaving, denoted by $\boldsymbol{x}^\frown \boldsymbol{y} \in \mathbb{R}^{2s}$, as follows: $(\boldsymbol{x}^\frown \boldsymbol{y})_{2i-1} = x_i$, $(\boldsymbol{x}^\frown \boldsymbol{y})_{2i} = y_i$ for all $i \in [s]$. The orthogonal vectors under finite-precision arithmetic can be:

**Lemma C.5** (Li et al., 2024). *For any $s \in \mathbb{N}^+$, let $\boldsymbol{q}_i = \mathrm{sbin}_s(i)^\frown 1_s$ and $\boldsymbol{k}_i = 2^{s+1} \cdot (\mathrm{sbin}_s(i)^\frown (-1_s))$ for all $i \in [2^s - 1]$, it holds that $\langle \boldsymbol{q}_i, \boldsymbol{k}_j \rangle_s = -B_s$ if $i \neq j$ and $\langle \boldsymbol{q}_i, \boldsymbol{k}_j \rangle_s = 0$ if $i = j$. Since $[\exp(-B_s)]_s \leq [2^{-s-1}]_s = 0$, it follows that $[\exp(\langle \boldsymbol{q}_i, \boldsymbol{k}_j \rangle_s)]_s = \mathbf{1}[i = j]$ for all $i, j \in [2^s - 1]$.*

### C.3.2   POSITION SELECTOR

**Lemma C.6.** *For any $m \in \mathbb{N}^+$ and $\boldsymbol{x} \in \mathbb{F}^s$, there exists a feedforward layer* $\mathrm{FF} : \mathbb{F}^{2s} \to \mathbb{F}^{2s}$, *for any $i \in [m]$, such that*

$$(\mathrm{id} + \mathrm{FF})((\boldsymbol{x}, \boldsymbol{e}_i)) = (\boldsymbol{x} \odot \boldsymbol{e}_i, \boldsymbol{e}_i). \tag{5}$$

*Proof.* Let the input be $\boldsymbol{z} = (\boldsymbol{x}, \boldsymbol{e}_i) \in \mathbb{F}^{2s}$. Set the weight $\boldsymbol{W}_1 \in \mathbb{F}^{s \times 2s}$ and bias $\boldsymbol{b} \in \mathbb{F}^s$ by

$$\boldsymbol{W}_1 = \begin{bmatrix} \boldsymbol{I}_s & -B_s \boldsymbol{I}_s \end{bmatrix}, \qquad \boldsymbol{b} = \boldsymbol{0}, \tag{6}$$

to have $\boldsymbol{W}_1 \boldsymbol{z} + \boldsymbol{b} = \boldsymbol{x} - B_s \boldsymbol{e}_i$. Applying the ReLU activation coordinate-wise gives

$$\mathrm{ReLU}(\boldsymbol{x} - B_s \boldsymbol{e}_i)_j = \begin{cases} 0, & j = i, \\ x_j, & j \neq i. \end{cases} \tag{7}$$

Hence,

$$\boldsymbol{h} := \mathrm{ReLU}(\boldsymbol{W}_1 \boldsymbol{z} + \boldsymbol{b}) = \boldsymbol{x} \odot (\boldsymbol{1} - \boldsymbol{e}_i). \tag{8}$$

Next, set the second linear layer $\boldsymbol{W}_2 \in \mathbb{F}^{2s \times s}$ by

$$\boldsymbol{W}_2 = \begin{bmatrix} -\boldsymbol{I}_s \\ \boldsymbol{0}_{s \times s} \end{bmatrix}. \tag{9}$$

Thus we have

$$\boldsymbol{z} + \boldsymbol{W}_2 \boldsymbol{h} = \begin{bmatrix} \boldsymbol{x} \\ \boldsymbol{e}_i \end{bmatrix} + \begin{bmatrix} -\boldsymbol{h} \\ \boldsymbol{0} \end{bmatrix} = \begin{bmatrix} \boldsymbol{x} \\ \boldsymbol{e}_i \end{bmatrix} + \begin{bmatrix} -\boldsymbol{x} \odot (\boldsymbol{1} - \boldsymbol{e}_i) \\ \boldsymbol{0} \end{bmatrix} = \begin{bmatrix} \boldsymbol{x} \odot \boldsymbol{e}_i \\ \boldsymbol{e}_i \end{bmatrix}. \tag{10}$$

$\square$

### C.3.3   FEEDFORWARD LAYERS

**Lemma C.7.** *Let $N \in \mathbb{N}^+$. For each $i \in [N]$, let $f_i : \Sigma^{\ell_i} \to \Sigma$ admit polynomially-efficient approximation by $\mathrm{FF}_i$. Then there exists a feedforward layer* $\mathrm{FF} : \mathbb{F}_s^{\sum_{i=1}^N \ell_i |\Sigma|} \to \mathbb{F}_s^{N|\Sigma|}$ *such that for every input tuple $\boldsymbol{x}_i = (x_1^{(i)}, \dots, x_{\ell_i}^{(i)}) \in \Sigma^{\ell_i}$,*

$$\mathrm{FF}\left( \Big\|_{i=1}^N \big(\boldsymbol{e}(x_1^{(i)}), \dots, \boldsymbol{e}(x_{\ell_i}^{(i)})\big) \right) = \Big\|_{i=1}^N \mathrm{FF}_i(\boldsymbol{x}_i), \tag{11}$$

*where $\|$ denotes concatenation.*

*Proof.* For each $i \in [N]$, by Definition C.4, there exist width $w_i \in \mathbb{N}$ and parameters

$$\boldsymbol{W}_1^{(i)} \in \mathbb{F}_s^{w_i \times \ell_i |\Sigma|}, \quad \boldsymbol{W}_2^{(i)} \in \mathbb{F}_s^{|\Sigma| \times w_i}, \quad \boldsymbol{b}^{(i)} \in \mathbb{F}_s^{w_i}$$

such that $\mathrm{FF}_i(\boldsymbol{x}_i) = \boldsymbol{W}_2^{(i)} \mathrm{ReLU}(\boldsymbol{W}_1^{(i)} \boldsymbol{e}(\boldsymbol{x}_i) + \boldsymbol{b}^{(i)})$.

Now, define block-diagonal matrices

$$\boldsymbol{W}_1 := \bigoplus_{i=1}^N \boldsymbol{W}_1^{(i)}, \qquad \boldsymbol{W}_2 := \bigoplus_{i=1}^N \boldsymbol{W}_2^{(i)}, \qquad \boldsymbol{b} := \bigoplus_{i=1}^N \boldsymbol{b}^{(i)}.$$

Then the single feedforward layer

$$\mathrm{FF}(\boldsymbol{x}) := \boldsymbol{W}_2 \mathrm{ReLU}(\boldsymbol{W}_1 \boldsymbol{x} + \boldsymbol{b})$$

applies each block independently to its corresponding input, yielding exactly $\mathrm{FF}(\boldsymbol{x}) = \Big\|_{i=1}^N \mathrm{FF}_i(\boldsymbol{x}_i)$.

$\square$

**Lemma C.8.** *Let $f : \Sigma^\ell \to \Sigma$ be a function that admits polynomially-efficient approximation. Then, there exist two feedforward layers*

$$\mathrm{FF}_1 : \mathbb{F}_s^{(1+\ell)|\Sigma|+1} \to \mathbb{F}_s^{(1+\ell)|\Sigma|+1}, \quad \mathrm{FF}_2 : \mathbb{F}_s^{(1+\ell)|\Sigma|+1} \to \mathbb{F}_s^{(1+\ell)|\Sigma|+1},$$

*such that, for every $\boldsymbol{x} = (x_1, \dots, x_\ell) \in \Sigma^\ell$ and $t \in \{0, 1\}$,*

$$(\mathrm{id} + \mathrm{FF}_2) \circ (\mathrm{id} + \mathrm{FF}_1)\big(\boldsymbol{0}, \, \boldsymbol{e}(x_1), \dots, \boldsymbol{e}(x_\ell), \, t\big) = \big(t \cdot \boldsymbol{e}(f(\boldsymbol{x})), \, \boldsymbol{e}(x_1), \dots, \boldsymbol{e}(x_\ell), \, t\big). \tag{12}$$

*Proof.* Let $n := |\Sigma|$ and $L := \ell n$. Write $\boldsymbol{u} = (\boldsymbol{e}(x_1), \dots, \boldsymbol{e}(x_\ell)) \in \{0,1\}^L$. By Definition C.4, there exist $w_f \in \mathbb{N}$, matrices $\boldsymbol{W}_1^{(f)} \in \mathbb{F}_s^{w_f \times L}$, $\boldsymbol{W}_2^{(f)} \in \mathbb{F}_s^{n \times w_f}$, and bias $\boldsymbol{b}^{(f)} \in \mathbb{F}_s^{w_f}$ such that

$$\mathrm{FF}_f(\boldsymbol{u}) := \boldsymbol{W}_2^{(f)} \mathrm{ReLU}(\boldsymbol{W}_1^{(f)}\boldsymbol{u} + \boldsymbol{b}^{(f)}), \quad \mathrm{FF}_f(\boldsymbol{u})_i = \begin{cases} \geq 1 - \delta & \text{if } \boldsymbol{e}(f(\boldsymbol{x})) = \boldsymbol{e}_i, \\ \leq \delta & \text{otherwise.} \end{cases} \tag{13}$$

Set the first layer as

$$\boldsymbol{W}_1^{(1)} = \begin{bmatrix} \boldsymbol{0} & \boldsymbol{W}_1^{(f)} & \boldsymbol{0}_{w_f \times 1} \\ \boldsymbol{0} & \boldsymbol{I}_L & \boldsymbol{0}_{L \times 1} \end{bmatrix}, \quad \boldsymbol{b}^{(1)} = \begin{bmatrix} \boldsymbol{b}^{(f)} \\ \boldsymbol{0}_L \end{bmatrix}, \quad \boldsymbol{W}_2^{(1)} = \begin{bmatrix} \boldsymbol{W}_2^{(f)} & \boldsymbol{0} \\ \boldsymbol{0}_{(L-n) \times w_f} & \boldsymbol{0} \\ \boldsymbol{0}_{1 \times w_f} & \boldsymbol{0} \end{bmatrix}, \tag{14}$$

and define $\mathrm{FF}_1(\boldsymbol{u}, t) := \boldsymbol{W}_2^{(1)} \mathrm{ReLU}\big(\boldsymbol{W}_1^{(1)}(\boldsymbol{u}, t) + \boldsymbol{b}^{(1)}\big)$. Then, it holds that

$$\mathrm{FF}_1(\boldsymbol{u}, t) = \boldsymbol{W}_2^{(1)} \mathrm{ReLU}\left(\begin{bmatrix} \boldsymbol{W}_1^{(f)}\boldsymbol{u} + \boldsymbol{b}^{(f)} \\ \boldsymbol{0} \end{bmatrix}\right) \tag{15}$$

$$= \begin{bmatrix} \boldsymbol{W}_2^{(f)}\mathrm{ReLU}(\boldsymbol{W}_1^{(f)}\boldsymbol{u} + \boldsymbol{b}^{(f)}) \\ \boldsymbol{0} \end{bmatrix} \tag{16}$$

$$= \begin{bmatrix} \mathrm{FF}_f(\boldsymbol{u}) \\ \boldsymbol{0} \end{bmatrix}. \tag{17}$$

Thus we have $(\mathrm{id} + \mathrm{FF}_1)(\boldsymbol{u}, t) = \big(\mathrm{FF}_f(\boldsymbol{u}), \, \boldsymbol{0}, \, t\big)$.

For the second layer, choose $\delta \leq \frac{1}{3}$ and $M \geq 1$ and set

$$\boldsymbol{W}_1^{(2)} = \begin{bmatrix} 2\boldsymbol{I}_n & M\boldsymbol{1}_n \\ 2\boldsymbol{I}_n & M\boldsymbol{1}_n \\ \boldsymbol{I}_n & 0 \\ -\boldsymbol{I}_n & 0 \end{bmatrix}, \; \boldsymbol{b}^{(2)} = \begin{bmatrix} -M\boldsymbol{1}_n \\ (1-M)\boldsymbol{1}_n \\ \boldsymbol{0}_n \\ \boldsymbol{0}_n \end{bmatrix}, \; \boldsymbol{W}_2^{(2)} = \begin{bmatrix} \boldsymbol{I}_n & -\boldsymbol{I}_n & -\boldsymbol{I}_n & \boldsymbol{I}_n \\ \boldsymbol{0}_{(L-n) \times n} & \boldsymbol{0} & \boldsymbol{0} & \boldsymbol{0} \\ \boldsymbol{0}_{1 \times n} & \boldsymbol{0} & \boldsymbol{0} & \boldsymbol{0} \end{bmatrix}, \tag{18}$$

and define $\mathrm{FF}_2(\boldsymbol{y}) := \boldsymbol{W}_2^{(1)} \mathrm{ReLU}\big(\boldsymbol{W}_1^{(1)}\boldsymbol{y} + \boldsymbol{b}^{(1)}\big)$. Then it holds that, for $\boldsymbol{z} := \mathrm{FF}_f(\boldsymbol{u})$,

$$\mathrm{FF}_2(\boldsymbol{z}, \boldsymbol{0}, t) = \boldsymbol{W}_2^{(2)} \begin{bmatrix} \mathrm{ReLU}\big(2\boldsymbol{z} + Mt\boldsymbol{1}_n - M\boldsymbol{1}_n\big) \\ \mathrm{ReLU}\big(2\boldsymbol{z} + Mt\boldsymbol{1}_n + (1-M)\boldsymbol{1}_n\big) \\ \mathrm{ReLU}(\boldsymbol{z}) \\ \mathrm{ReLU}(-\boldsymbol{z}) \end{bmatrix} \tag{19}$$

$$= \boldsymbol{W}_2^{(2)} \begin{bmatrix} \mathrm{ReLU}\big(2\boldsymbol{z} + M(t-1)\boldsymbol{1}_n\big) \\ \mathrm{ReLU}\big(2\boldsymbol{z} + 1 + M(t-1)\boldsymbol{1}_n\big) \\ \boldsymbol{z} \\ \boldsymbol{0} \end{bmatrix} \tag{20}$$

$$= \begin{bmatrix} \mathrm{ReLU}\big(\boldsymbol{z} - \delta + M(t-1)\big) - \mathrm{ReLU}\big(\boldsymbol{z} - 1 + M(t-1)\big) - \boldsymbol{z} \\ \boldsymbol{0} \\ 0 \end{bmatrix}, \tag{21}$$

where it satisfies that

$$\mathrm{ReLU}\big(\boldsymbol{z} - \delta + M(t-1)\big) - \mathrm{ReLU}\big(\boldsymbol{z} - \delta - 1 + M(t-1)\big) = \begin{cases} \boldsymbol{e}(f(\boldsymbol{x})) & \text{if } t = 1, \\ \boldsymbol{0} & \text{if } t = 0. \end{cases} \tag{22}$$

Therefore, the composition satisfies $(\mathrm{id} + \mathrm{FF}_2) \circ (\mathrm{id} + \mathrm{FF}_1)\big(\boldsymbol{u}, t\big) = (t \cdot \boldsymbol{e}(f(\boldsymbol{x})), \boldsymbol{0}_L, t)$. $\qquad\square$

## C.4 PROOF FOR THEOREM 3.5

*Proof.* Let $G_n = (V_n, E_n)$ be a computation graph, where $\mathcal{F} = \{f_1, f_2, \dots, f_{|\mathcal{F}|}\}$. Each node $v \in V_n$ is labeled by a one-hot vector $\boldsymbol{e}(v) \in \{0,1\}^{|\mathcal{F}|}$ indicating the function assigned to $v$ from the finite set $\mathcal{F}$. Let $v_1, v_2, \dots, v_{|V_n|}$ denote a fixed topological ordering of $V_n$, with inputs appearing first and outputs last. For each function $f_i \in \mathcal{F}$, let $C_{f_i}(n) := \max\{ |\mathrm{pred}(v)| : v \in V_n, \, \boldsymbol{e}(v) = \boldsymbol{e}_i \}$. and define $C_{\mathrm{sum}}(n) := \sum_{f \in \mathcal{F}} C_f(n)$, $C_{\max}(n) := \max_{f \in \mathcal{F}} C_f(n)$.

Let the precision be $s(n) = C \cdot \lceil \log_2 n \rceil$ where $C \in \mathbb{N}$ is a sufficiently large integer such that $2^{s(n)} \geq n^C$ exceeds the maximum polynomial step bound under consideration. We denote by $\mathrm{pred}(v_i) \in \mathbb{F}_{s(n)}^{C_{\max}(n)}$ the vector of predecessor indices of node $v_i$; that is, if $v_i$ has $d \leq C_{\max}(n)$ incoming edges from nodes $v_{j_1}, \ldots, v_{j_d}$, then $\mathrm{pred}(v_i) = (j_1, \ldots, j_d, \mathbf{0})$, where zeros are used for padding so that the length is exactly $C_{\max}(n)$.

Let the vocabulary be $\mathcal{V} = \Sigma$. At decoding step $k$, the model has access to the concatenated sequence

$$(x_1, x_2, \ldots, x_n, y_1, y_2, \ldots, y_k) \in \Sigma^{n+k}, \tag{23}$$

where $x = (x_1, \ldots, x_n) \in \Sigma^n$ denotes the input, and $y_i$ is the token generated at the $i$-th CoT step. For each node $v_j$, let $v_j(x)$ denote its value on input $x$. We assume that every intermediate output satisfies $y_i = v_{n+i}(x)$. Under this assumption, we prove by induction that the model generates the next token correctly, i.e., $y_{k+1} = v_{n+k+1}(x)$.

**Embedding**  The embedding at position $i \in [n+k]$, denoted by $\boldsymbol{h}_i^{(0)} \in \mathbb{F}_{s(n)}^m$, where $m := |\Sigma| + |\mathcal{F}| + (1 + C_{\max}(n))s(n) + |\Sigma|C_{\mathrm{sum}}(n)$ is defined as

$$\boldsymbol{h}_i^{(0)} = \left( \boldsymbol{e}(v_i(x)), \, \boldsymbol{e}(v_{i+1}), \, \mathsf{sbin}_{s(n)}(i), \, \mathbf{sbinpred}_{s(n)}(v_{i+1}), \, \mathbf{0}_{|\Sigma|C_{\mathrm{sum}}(n)} \right), \tag{24}$$

where $\boldsymbol{e} \colon \Sigma \to \{0,1\}^{|\Sigma|}$ denote the one-hot encoding of the symbol and, $\mathbf{sbinpred}_{s(n)}(v) \in \mathbb{F}_{s(n)}^{C_{\max}(n) \cdot s(n)}$ encodes the binary representations of the predecessor indices:

$$\mathbf{sbinpred}_{s(n)}(v) = \left( \mathsf{sbin}_{s(n)}(\mathrm{pred}(v_i)_0), \, \ldots, \, \mathsf{sbin}_{s(n)}(\mathrm{pred}(v_i)_{C_{\max}(n)}) \right). \tag{25}$$

This embedding is constructed, for $z \in \Sigma$, as

$$\mathbf{WE}(z) = (\boldsymbol{e}(z), \mathbf{0}), \quad \mathbf{PE}(i) = \left( \mathbf{0}, \, \boldsymbol{e}(v_{i+1}), \, \mathsf{sbin}_{s(n)}(i), \, \mathbf{sbinpred}_{s(n)}(v_{i+1}), \, \mathbf{0} \right). \tag{26}$$

**Attention layer**  The first attention layer consists of $C_{\max(n)}$ heads. The $h$-th head is configured to attend to the position corresponding to the $h$-th predecessor. Specifically, for each position $i$ and head $h \in [C_{\max(n)}]$, the attention vectors are defined as:

$$\boldsymbol{q}_{i,h} = \mathsf{sbin}_{s(n)}(\mathrm{pred}(v_i)_h) ^\frown 1_{s(n)}, \tag{27}$$

$$\boldsymbol{k}_{i,h} = 2^{s(n)+1} \cdot \mathsf{sbin}_{s(n)}(i) ^\frown (-1_{s(n)}), \tag{28}$$

$$\mathbf{v}_{i,h} = \boldsymbol{e}(v_i(x)), \tag{29}$$

where vectors of different lengths are zero-padded to match the dimension. By Lemma C.5, each attention head of the last position $i = n + k$ retrieves the predecessor's value of $v_{n+k}$

$$\boldsymbol{a}_{n+k,h} = \boldsymbol{e}\left( v_{\mathrm{pred}(v_{n+k})_h}(x) \right) \tag{30}$$

With an appropriate output projection $\boldsymbol{O}$ such that

$$\boldsymbol{O}(\boldsymbol{a}_{i,1}, \ldots, \boldsymbol{a}_{i,H}) = (\mathbf{0}, \, \boldsymbol{e}(v_{\mathrm{pred}(v_{n+k})_0}(x)), \, \ldots, \, \boldsymbol{e}(v_{\mathrm{pred}(v_{n+k})_{C_{\max(n)}}}(x)), \, \mathbf{0}), \tag{31}$$

the hidden state at position $n + k$ after the attention layer is given by

$$\boldsymbol{h}_{n+k}^{(0.5)} = \Big( \boldsymbol{e}(v_{n+k}(x)), \, \boldsymbol{e}(v_{n+k+1}), \, \mathsf{sbin}_{s(n)}(n+k), \, \mathbf{sbinpred}_{s(n)}(v_{n+k+1}), \tag{32}$$

$$\underbrace{\boldsymbol{e}(v_{\mathrm{pred}(v_{n+k})_0}(x)), \, \ldots, \, \boldsymbol{e}(v_{\mathrm{pred}(v_{n+k})_{C_{\max(n)}}}(x))}_{\text{updated}}, \, \mathbf{0}_{C_{\mathrm{sum}}(n)|\Sigma|} \Big). \tag{33}$$

The second and third attention layers are disabled (i.e., all attention weights are set to zero).

**Feed-forward layer**  By Lemma C.7, a single feed-forward layer can approximate multiple functions by partitioning the input into blocks. The first feed-forward layer then places the arguments, gathered by attention, into the correct positions. By Lemma C.6, where the vector $e_i$ therein corresponds to $\mathbf{1}_{|\Sigma|}$ here, the hidden state at the last position, denoted by $h^{(1)}_{n+k}$, becomes

$$\left((h^{(0.5)}_{n+k})_{1:r}, \big\|^{|\mathcal{F}|}_{j=1}\big(e(v_{\text{pred}(v_{n+k})_1}(x)) \cdot 1, \ldots, e(v_{\text{pred}(v_{n+k})_{C_j(n)}}(x)) \cdot 1\big)\right), \tag{34}$$

where $r = |\Sigma| + |\mathcal{F}| + (1 + C_{\max})(n)s(n)$.

By Assumption 3.4 and Lemmas C.7 and C.8, there exist feed-forward layers $\text{FF}_2, \text{FF}_3 : \mathbb{F}^m_{s(n)} \to \mathbb{F}^m_{s(n)}$ such that, for every input tuple $x_j := (x^{(j)}_1, \ldots, x^{(j)}_{C_j(n)}) \in \Sigma^{C_j(n)}$ for $j \in [|\mathcal{F}|]$ and every $t \in \{0,1\}^{|\mathcal{F}|}$, the composition $\mathcal{FF} := (\text{id} + \text{FF}_3) \circ (\text{id} + \text{FF}_2)$ satisfies

$$\mathcal{FF}\left(*, t, *, \big\|^{|\mathcal{F}|}_{j=1}(e(x^{(j)}_1), \ldots, e(x^{(j)}_{C_j(n)}))\right) = \left(\sum_j^{|\mathcal{F}|} t_j \cdot e(f_j(x_j)), *\right), \tag{35}$$

zwhere $*$ denotes an unspecified vector. Since the second and third attention layers are disabled, after the third layer, applying the second and third feed-forward layers $\text{FF}_2, \text{FF}_3$, the hidden state becomes

$$h^{(3)}_{n+k} = \mathcal{FF}\left(h^{(1)}_{n+k}\right) \tag{36}$$

$$= \mathcal{FF}\left(*, e(v_{n+k+1}), *, \big\|^{|\mathcal{F}|}_{j=1}\big(e(v_{\text{pred}(v_{n+k})_1}(x)), \ldots, e(v_{\text{pred}(v_{n+k})_{C_j(n)}}(x))\big)\right) \tag{37}$$

$$= \left(\sum_{j=1}^{|\mathcal{F}|} e(v_{n+k+1})_j \cdot e\Big(f_j\big(e(v_{\text{pred}(v_{n+k})_1}(x)), \ldots, e(v_{\text{pred}(v_{n+k})_{C_j(n)}}(x))\big)\Big), *\right) \tag{38}$$

$$= \left(e\Big(f_l\big(e(v_{\text{pred}(v_{n+k})_1}(x)), \ldots, e(v_{\text{pred}(v_{n+k})_{C_l(n)}}(x))\big)\Big), *\right), \quad \text{where} \quad e(v_{n+k+1}) = e_l \tag{39}$$

$$= \left(e\big(v_{n+k+1}(x)\big), *\right). \tag{40}$$

**Output layer**  The final output is given by

$$h_{n+k} = \mathbf{OUT}(h^{(3)}_{n+k}) = \begin{bmatrix} I_{|\Sigma|} & 0 \end{bmatrix} h^{(3)}_{n+k} = e(v_{n+k+1}(x)). \tag{41}$$

The decoding function then outputs the symbol corresponding to the maximum score,

$$y_{n+k+1} = \text{Dec}(h_{n+k}) = \arg\max_{j \in [|\Sigma|]} e(v_{n+k+1}(x)) = v_{n+k+1}(x). \tag{42}$$

By induction on $k$, the model computes the values at all nodes in topological order. The parameter size of the model is determined by the requirements of the feedforward layers, $O(\text{ff\_param}(G_n))$. While the dimensions and heads of the attention layers depend on $C_{\max}(n)$, which is precisely what is already required for the feedforward layers to approximate the target functions. □

## C.5  Proof for Theorem 3.7

*Proof.*  We construct a model in which (1) the attention layer aggregates the inputs, and (2) the looped feed-forward layer performs the computation of all nodes in parallel, as illustrated in Figure 5.

We first show that a feedforward layer followed by an attention layer can copy all input tokens to the each position, as shown below. Assume, given an input sequence $x = (x_1, x_2, \ldots, x_n) \in \Sigma^n$ and one-hot encoding $e : \Sigma \to \{0,1\}^{|\Sigma|}$. Assume each input at position $i \in [n]$ is embedded as

$$h_i = \big(\mathbf{0}, \ e(x_i), \ e_i\big) \in \{0,1\}^{n|\Sigma|+|\Sigma|+n}, \tag{43}$$

where $e_i \in \{0,1\}^n$. By Lemma C.6, when substituting $x = (e(x_1), \ldots, e(x_n))$ into the lemma, there exists a feed-forward layer $\text{FF}_1$ such that

$$(\text{id} + \text{FF}_1)(h_i) = \big((e_i)_1 \cdot e(x_i), \ (e_i)_2 \cdot e(x_i), \ \ldots, \ (e_i)_n \cdot e(x_i), \ e(x_i), \ e_i\big). \tag{44}$$

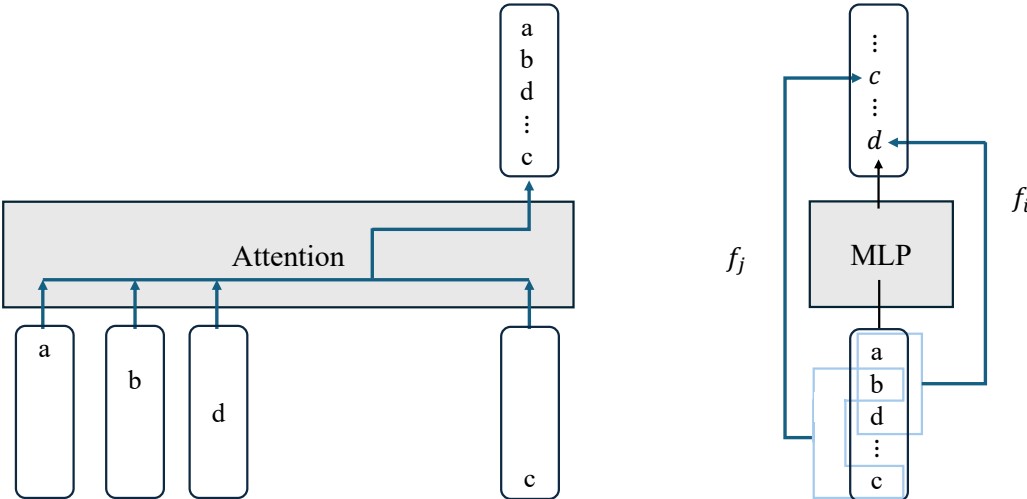

Figure 5: Illustration of the role of the attention and feedforward layers in looped TFs for evaluating DAGs: the attention layer uniformly attends to and aggregates all inputs at each position, while the feedforward layer simultaneously simulates the functions of all DAG nodes in parallel.

To aggregate all positions via uniform attention, we use a single-head attention layer with:

$$\mathbf{q}_i = \boldsymbol{k}_i = \mathbf{1}_{n|\Sigma|}, \quad \mathbf{v}_i = n\big((\boldsymbol{e}_i)_1 \cdot \boldsymbol{e}(x_i),\ (\boldsymbol{e}_i)_2 \cdot \boldsymbol{e}(x_i),\ \ldots,\ (\boldsymbol{e}_i)_n \cdot \boldsymbol{e}(x_i)\big) \quad \text{for all } i \in [n], \quad (45)$$

with an appropriate output projection, the output of the attention layer, at position $i$, becomes

$$\frac{1}{n}\sum_{j=1}^{n} 1 \cdot n\boldsymbol{h}_j \;=\; \Big(\sum_{j=1}^{n}(\boldsymbol{e}_j)_1\, \boldsymbol{e}(x_j),\ \sum_{j=1}^{n}(\boldsymbol{e}_j)_2\, \boldsymbol{e}(x_j),\ \ldots,\ \sum_{j=1}^{n}(\boldsymbol{e}_j)_n\, \boldsymbol{e}(x_j)\Big) \qquad (46)$$

$$= \big(\boldsymbol{e}(x_1),\ \boldsymbol{e}(x_2),\ \ldots,\ \boldsymbol{e}(x_n)\big). \qquad (47)$$

Then, we show that the feed-forward layer can encode the entire computation graph into its weights and simulate all nodes simultaneously. Let the flag vector for each node be $(t_1, \ldots, t_N) \in \{0,1\}^N$, where $N := |V_n|$. By Lemmas C.7 and C.8, there exist feed-forward layers $\mathrm{FF}_2, \mathrm{FF}_3 : \mathbb{F}_s^{N(|\Sigma|+1)} \to \mathbb{F}_s^{N(|\Sigma|+1)}$ such that, for the input vector $(z_1, \ldots, z_N) \in \Sigma^N$.,

$$\mathcal{FF}\Big(\boldsymbol{e}(z_1), t_1, \ldots, \boldsymbol{e}(z_N(x)), t_N\Big) = \big\|_{i=1}^{N}\Big(t_i \cdot \boldsymbol{e}\big(f_{v_i}(\boldsymbol{z}^{(i)})\big),\ \frac{1}{m_i}\sum_{j=1}^{m_i} t_{p_{i,j}}\Big), \qquad (48)$$

where $\mathcal{FF} := (\mathrm{id} + \mathrm{FF}_3) \circ (\mathrm{id} + \mathrm{FF}_2)$. Here, $f_{v_i}$ denotes the function associated with node $v_i$, and $p_{i,1}, \ldots, p_{i,m_i}$ denote the indices of the predecessor nodes of $v_i$, and $\boldsymbol{z}^{(i)} = (z_{p_{i,1}}, \ldots, z_{p_{i,m_i}})$ denotes their values. The last term $\frac{1}{m_i}\sum_{j=1}^{m_i} t_{p_{i,j}}$ can be obtained using a linear layer.

For the $k$-th loop, assume by induction that the hidden state is

$$\boldsymbol{h}(k) := \big(t_{1,k-1} \cdot \boldsymbol{e}(v_1(x)),\ t_{1,k},\ t_{2,k-1} \cdot \boldsymbol{e}(v_2(x)),\ t_{2,k},\ \ldots,\ t_{N,k-1} \cdot \boldsymbol{e}(v_N(x)),\ t_{N,k}\big), \qquad (49)$$

where $v_i(x) \in \Sigma$ denotes the value computed by node $v_i$ given the input $x$, and $t_{i,k} \in \{0,1\}$ indicates whether node $v_i$ lies within depth at most $k$. Under this assumption, it holds that

$$\mathcal{FF}(\boldsymbol{h}(k)) \;=\; \big\|_{i=1}^{N}\Big(t_{i,k} \cdot \boldsymbol{e}\big(f_{v_i}(v_{p_{i,1}}(x), v_{p_{i,2}}(x), \ldots, v_{p_{i,m_i}}(x))\big),\ \frac{1}{m_i}\sum_{j=1}^{m_i} t_{(p_{i,j},k)}\Big), \qquad (50)$$

$$= \big\|_{i=1}^{N}\Big(t_{i,k} \cdot \boldsymbol{e}\big(v_i(x)\big),\ t_{i,k+1}\Big) \;=\; \boldsymbol{h}(k+1). \qquad (51)$$

To extract the output node corresponding to each position denoted by $o_i$, in the final loop iteration, by Lemma C.6, there exists a feedforward layer $\mathrm{FF}_4$ such that

$$(\mathrm{id} + \mathrm{FF}_4)(\boldsymbol{h}(k), \boldsymbol{e}_{o_i}) = \big(t_{o_i,k} \cdot \boldsymbol{e}(v_{o_i}(x)),\ *\big). \qquad (52)$$

**Summary** We construct the looped model as follows. Each input token $x_i \in \Sigma$ at position $i \in [n]$ is embedded as $\boldsymbol{h}_i^{(0)} \in \{0,1\}^{2n+(1+|\Sigma|)N+|\Sigma|}$:

$$\boldsymbol{h}_i^{(0)} = \big( \boldsymbol{e}_i,\ \boldsymbol{e}_{o_i},\ \boldsymbol{e}(x_i),\ 0,\ \boldsymbol{e}(x_i),\ 0,\ \ldots,\ \boldsymbol{e}(x_i),\ 0,\ \boldsymbol{0}_{(1+|\Sigma|)(N-n)+|\Sigma|} \big). \qquad (53)$$

The first attention layer is an identity map, while the first feedforward layers compute

$$\boldsymbol{h}_i^{(1)} = \big( \boldsymbol{e}_i,\ \boldsymbol{e}_{o_i},\ (\boldsymbol{e}_i)_1 \cdot \boldsymbol{e}(x_i),\ 0,\ (\boldsymbol{e}_i)_2 \cdot \boldsymbol{e}(x_i),\ 0,\ \ldots,\ (\boldsymbol{e}_i)_n \cdot \boldsymbol{e}(x_i),\ 0,\ \boldsymbol{0}_{(1+|\Sigma|)(N-n)+|\Sigma|} \big). \qquad (54)$$

The second attention layer uniformly gathers all positions and appends a constant 1 to each block:

$$\boldsymbol{h}_i^{(1.5)} = \big( \boldsymbol{e}_i,\ \boldsymbol{e}_{o_i},\ \boldsymbol{e}(x_1),\ 1,\ \boldsymbol{e}(x_2),\ 1,\ \ldots,\ \boldsymbol{e}(x_n),\ 1,\ \boldsymbol{0}_{(1+|\Sigma|)(N-n)+|\Sigma|} \big) \qquad (55)$$

$$= \big( \boldsymbol{e}_i,\ \boldsymbol{e}_{o_i},\ \boldsymbol{h}(0),\ \boldsymbol{0}_{|\Sigma|} \big). \qquad (56)$$

We now proceed by induction. Assume that at the $k$-th iteration, the output after the second attention layer at position $i$ is

$$\boldsymbol{h}_{k,i}^{(1.5)} = \big( \boldsymbol{e}_i,\ \boldsymbol{e}_{o_i},\ \boldsymbol{h}(k),\ t_{o_i,k-1} \cdot \boldsymbol{e}(v_{o_i}(x)) \big), \qquad (57)$$

where $t_{o_i,k-1} := \big( \boldsymbol{h}_{k-1,i}^{(1.5)} \big)_{2n+(1+|\Sigma|)(o_i-1)}$ denotes the indicator flag showing whether the $o_i$-th node has been reached, i.e., whether it lies at depth $k-1$.

After passing through the second feedforward layer, the third attention layer with all weights set to zero, and the third feedforward layer, the hidden state updates to

$$\boldsymbol{h}_{k,i}^{(3)} = \big( \boldsymbol{e}_i,\ \boldsymbol{e}_{o_i},\ \boldsymbol{h}(k+1),\ t_{o_i,k-1} \cdot \boldsymbol{e}(v_{o_i}(x)) \big). \qquad (58)$$

After the fourth feedforward layer, the hidden state becomes

$$\boldsymbol{h}_{k,i}^{(4)} = \big( \boldsymbol{e}_i,\ \boldsymbol{e}_{o_i},\ \boldsymbol{h}(k+1),\ t_{o_i,k} \cdot \boldsymbol{e}(v_{o_i}(x)) \big). \qquad (59)$$

By induction, after the final iteration of depth $\mathrm{depth}(G_n)$ of the computation graph $G_n$, we obtain

$$\boldsymbol{h}_{\mathrm{depth}(G_n),i}^{(4)} = \big( *,\ t_{o_i,\mathrm{depth}(G_n)} \cdot \boldsymbol{e}(v_{o_i}(x)) \big) = \big( *,\ \boldsymbol{e}(v_{o_i}(x)) \big). \qquad (60)$$

The final output is given by

$$z_i = \mathbf{OUT}(\boldsymbol{h}_{\mathrm{depth}(G_n),i}^{(4)}) = \begin{bmatrix} \boldsymbol{0} & \boldsymbol{I}_{|\Sigma|} \end{bmatrix} (\boldsymbol{h}_{\mathrm{depth}(G_n),i}^{(4)}) = \boldsymbol{e}(v_{o_i}(x)). \qquad (61)$$

The decoding function then selects the symbol corresponding to the maximum score:

$$y_i = \mathrm{Dec}(z_i) = \arg \max_{j \in [|\Sigma|]} \big( \boldsymbol{e}(v_{o_i}(x)) \big)_j = v_{o_i}(x). \qquad (62)$$

The parameter size grows proportionally with the input dimension $\mathrm{size}(G_n)$, since the function must be approximated along each dimension. Therefore, it can be bounded by $O(\mathrm{ff\_param}(G_n) \cdot \mathrm{size}(G_n))$. $\qquad \square$

**Discussion:** One might object that our proof, compressing the entire length $n$ input to a single position and then applying an arbitrary feed-forward computation, deviates from standard Transformer architecture. An alternative is to distribute computation across positions, as shown by Sanford et al. (2024b), which can reduce the embedding dimension. We nevertheless adopt the single-position construction to isolate the essence of looped models: attention serves solely to aggregate information, a single layer is sufficient, while the feed-forward network performs the computation in latent space, which is more expressive than computations in the language space. This perspective aligns with expressivity results for looped ReLU networks (Liang et al., 2024).

## C.6 Proof for Theorem 3.12

For the upper bound $\mathsf{Loop}[\log^k n, \mathsf{poly}(n), 1 \text{ (resp. } \log n)] \subseteq \mathsf{AC}^k$ (resp. $\mathsf{TC}^k$), we follow the argument of Li et al. (2024). Their key observation is that a restricted form of automaton can model iterative computation under constant precision: the rounding operation preserves monotonicity, and constant precision yields counter-free, restricted state spaces, which are therefore computable by $\mathsf{AC}^0$ circuits. In the case of polynomial precision, prefix summation can be simulated by $\mathsf{TC}^0$, which also allows the detection and correction of rounding in fixed-point arithmetic.

**Theorem C.9** (Li et al., 2024). *For $s(n) \in \mathsf{poly}(n)$, $\mathrm{sum}_{s(n)} : (\mathbb{F}_{s(n)})^n \to \mathbb{F}_{s(n)}$ is computable by $\mathsf{TC}^0$ circuits, and by $\mathsf{AC}^0$ circuits when $s(n)$ is constant.*

It has also been shown that gates can be efficiently simulated by feedforward layers.

**Lemma C.10** (Li et al., 2024). *Unbounded-fanin $\mathsf{AND}, \mathsf{OR}$ (resp. MAJORITY) $: \{0,1\}^n \to \{0,1\}$ can be simulated by a two-layer feedforward ReLU network with constant (resp. $\log n$) bits of precision constant hidden dimension and additional $n$ constant inputs of value $1$.*

*Proof for Theorem 3.12.* $\mathsf{Loop}[\log^k n, \mathsf{poly}(n), 1 \text{ (resp. } \log n)] \subseteq \mathsf{AC}^k$ (resp. $\mathsf{TC}^k$) : In Transformers, constant-depth computation is defined by Summation with Iterative Rounding (see Definition C.3), which by Theorem C.9 can be simulated in $\mathsf{AC}^0$ (resp. $\mathsf{TC}^0$). A looped TF simply stacks these computations vertically through iteration, and thus the result follows.

$\mathsf{AC}^k$ (resp. $\mathsf{TC}^k$) $\subseteq \mathsf{Loop}[\log^k n, \mathsf{poly}(n), 1 \text{ (resp. } \log n)]$ : Since Boolean circuits are DAGs, the claim follows directly from Theorem 3.11 together with Lemma C.10. $\qquad\square$

# D Deferred Proofs for Section 4

## D.1 Preliminaries

In this section, we state the deferred definitions and present the lemma.

### D.1.1 Models of Computation

To analyze the distinction between CoT and latent thought in the context of next-token prediction, we formalize CoT as a process that, given the prefix $y_{<i} = (y_1, y_2, \ldots, y_{i-1})$, generates the next token $y_i$ via a stochastic reasoning step, as follows.

**Definition D.1** (CoT with stochastic decoding). Let $\mathcal{V}$ be a vocabulary. Given an input $x \in \mathcal{V}^*$, CoT stochastically generates a sequence of output blocks of the form

$$\Big( r_1, e, y_1, e', r_2, e, y_2, e', \cdots r_m, e, y_m, e' \Big)$$

where each $r_i \in \mathcal{V}^*$ is a reasoning path and $e, e' \in \mathcal{V}$ are special delimiter tokens, yielding the final output string $y_1 \ldots y_m$. Generation proceeds *autoregressively* via *next-token prediction*: for each $i \in [m]$, the model generates a reasoning step $r_i$ followed by an output token $y_i$, conditioned on the input $x$, previous outputs $y_{<i}$, and prior reasoning steps $r_{<i}$.

For looped models, we use the last output of looped TF as the next token.

**Definition D.2** (looped TF with stochastic decoding). Let $f$ be a looped TF with vocabulary $\Sigma$ and loop iteration bound $T(n) \in \mathsf{poly}(n)$. Given an input $x \in \Sigma^*$ and prefix $y_{<k} = (y_1, \ldots, y_{k-1})$, its hidden representation at the last position after $T(|x|)$ iterations is

$$h(x, y_{\leq k-1}) := f^{T(|x|)}(x, y_{\leq k-1})_{n+k-1},$$

A stochastic decoding function $\mathrm{Dec} : \mathbb{F}^d \to \Delta(\Sigma)$ maps the hidden state at each position to a probability distribution over tokens. The autoregressive generation distribution of the model is then

$$\Pr(y \mid x) := \prod_{k=1}^{m} \mathrm{Dec}\big(h(x, y_{<k})\big)(y_k),$$

for every $y = (y_1, \ldots, y_m) \in \Sigma^*$.

### D.1.2 SELF-CONSISTENCY ENHANCES RELIABILITY

The lemma shows that the success probability can be improved by repeating the trial multiple times.

**Lemma D.3** (Error reduction). *Suppose there is a randomized estimator $A$ for a nonnegative function $f$ such that for every input $x$, every accuracy parameter $\varepsilon > 0$, and some constant $\gamma \in (0, 1/2)$,*

$$\Pr[(1 - \varepsilon)f(x) \leq A(x, \varepsilon) \leq (1 + \varepsilon)f(x)] \geq \tfrac{1}{2} + \gamma, \tag{63}$$

*and the running time of $A$ is polynomial in $|x|$ and $1/\varepsilon$. Then, for every confidence parameter $\delta \in (0, 1)$, there is an FPRAS for $f$ outputs $\widehat{f}(x)$ satisfying*

$$\Pr\Big[(1 - \varepsilon)f(x) \leq \widehat{f}(x) \leq (1 + \varepsilon)f(x)\Big] \geq 1 - \delta. \tag{64}$$

*Proof.* Fix $x, \varepsilon, \delta$. Run $A$ independently $k$ times with fresh randomness and the same $(x, \varepsilon)$, obtaining estimates $\widehat{f}^{(1)}, \ldots, \widehat{f}^{(k)}$. Output the median

$$\widehat{f}_{\mathrm{med}} := \operatorname{median}\Big\{\widehat{f}^{(1)}, \ldots, \widehat{f}^{(k)}\Big\}. \tag{65}$$

For each run $i \in \{1, \ldots, k\}$, define the Bernoulli indicator

$$Y_i := \mathbf{1}\Big[(1 - \varepsilon)f(x) \leq \widehat{f}^{(i)} \leq (1 + \varepsilon)f(x)\Big] \in \{0, 1\}, \tag{66}$$

so that $p := \Pr[Y_i = 1] \geq \tfrac{1}{2} + \gamma$ by assumption, and let $S := \sum_{i=1}^{k} Y_i$, so $\mathbb{E}[S] = pk$. If at least half the runs are successful (i.e., $S \geq \lceil k/2 \rceil$), then at least $\lceil k/2 \rceil$ samples lie inside the interval $\mathcal{I} = [(1 - \varepsilon)f(x), (1 + \varepsilon)f(x)]$, hence the median must also lie in $\mathcal{I}$. Therefore

$$\Pr\Big[\widehat{f}_{\mathrm{med}} \notin \mathcal{I}\Big] \leq \Pr\Big[S < \frac{k}{2}\Big]. \tag{67}$$

Since the $Y_i$ are independent in $[0, 1]$, Hoeffding's inequality gives

$$\Pr\left[S \leq \frac{k}{2}\right] = \Pr\left[\frac{S}{k} - p \leq -\Big(p - \frac{1}{2}\Big)\right] \leq \exp\big(-2k\,(p - \tfrac{1}{2})^2\big) \leq \exp\big(-2k\,\gamma^2\big). \tag{68}$$

Choose $k \geq \frac{1}{2\gamma^2} \ln\frac{1}{\delta}$, then $\Pr[\widehat{f}_{\mathrm{med}} \notin \mathcal{I}] \leq \delta$. We invoke $A$ independently $k = O\big(\gamma^{-2} \log(1/\delta)\big)$ times and take a median, Hence the total running time is polynomial in $|x|$, $1/\varepsilon$, and $\log(1/\delta)$. $\square$

**Lemma D.4.** *For $m \geq 1$ and $-m \leq \varepsilon \leq \sqrt{m}$, it holds that*

$$e^{\varepsilon - \frac{1}{2}} \leq \Big(1 + \frac{\varepsilon}{m}\Big)^m \leq e^{\varepsilon}. \tag{69}$$

*Proof.* For $u > -1$, the elementary bounds $\ln(1+u) \leq u$ and $\ln(1+u) \geq u - \frac{u^2}{2}$ hold. Substituting $u = \varepsilon/m$ and multiplying by $m$ gives

$$m\ln\Big(1 + \frac{\varepsilon}{m}\Big) \leq \varepsilon, \qquad m\ln\Big(1 + \frac{\varepsilon}{m}\Big) \geq \varepsilon - \frac{\varepsilon^2}{2m}. \tag{70}$$

This yields, respectively,

$$(1 + \varepsilon/m)^m \leq e^{\varepsilon}, \qquad (1 + \varepsilon/m)^m \geq \exp\big(\varepsilon - \varepsilon^2/(2m)\big). \tag{71}$$

If $\varepsilon^2 \leq m$, then $\varepsilon^2/(2m) \leq \frac{1}{2}$, thus we have $(1 + \varepsilon/m)^m \geq e^{\varepsilon - 1/2}$. $\square$

### D.1.3 Self-Reducibility

**Definition D.5** (Schnorr, 1976). A relation $R \subseteq \Sigma^* \times \Sigma^*$ is *self-reducible* if:

1. There exists a polynomial-time computable function $g \in \Sigma^* \to \mathbb{N}$ s.t., $(x, y) \in R \Rightarrow |y| = g(x)$;

2. There exists a polynomial-time Turing machine that decides membership in $R$.

3. There exist polynomial-time computable functions $\psi \in \Sigma^* \times \Sigma^* \to \Sigma^*$ and $\sigma \in \Sigma^* \to \mathbb{N}$ s.t.

$$\sigma(x) = O(\log |x|), \tag{72}$$

$$g(x) > 0 \Rightarrow \sigma(x) > 0 \quad \forall x \in \Sigma^*, \tag{73}$$

$$|\psi(x, w)| \leq |x| \quad \forall x, w \in \Sigma^*, \tag{74}$$

and such that, for all $x \in \Sigma^*$, $y = y_1 \dots y_n \in \Sigma^*$,

$$\langle x, y_1, \dots, y_n \rangle \in R \iff \langle \psi(x, y_1 \dots y_{\sigma(x)}), y_{\sigma(x)+1}, \dots, y_n \rangle \in R. \tag{75}$$

For example, SAT is self-reducible: by fixing a prefix of variables and applying the reduction map $\psi$, the problem is simplified to a smaller instance whose solutions extend the chosen prefix. Consider the Boolean formula $F = (x_1 \vee x_2) \wedge (\neg x_1 \vee x_3) \wedge (\neg x_2 \vee \neg x_3)$, and suppose we fix the first variable to $x_1 = 1$. The residual instance is obtained by applying $\psi(F, (1))$, which substitutes $x_1 = 1$ and simplifies the formula by deleting satisfied clauses and removing falsified literals: $\psi(F, (1)) = (x_3) \wedge (\neg x_2 \vee \neg x_3)$. The unit clause $(x_3)$ forces $x_3 = 1$, which in turn simplifies $(\neg x_2 \vee \neg x_3)$ to $\neg x_2$, yielding $x_2 = 0$. Hence the unique residual assignment is $(x_2, x_3) = (0, 1)$, and together with the prefix $x_1 = 1$, we obtain the satisfying assignment $(x_1, x_2, x_3) = (1, 0, 1)$.

For self-reducible relations, the existence of an FPRAS and an FPAUS is equivalent.

**Theorem D.6** (Jerrum et al., 1986). *Let $R$ be a self-reducible relation. There exists an FPRAS for approximating $|R(x)|$ if and only if there exists an FPAUS for sampling uniformly from $R(x)$.*

## D.2 Proof for Proposition 4.8

The basic strategy follows the reduction from almost uniform generation to approximate counting (Jerrum et al., 1986).

*Proof.* Fix target parameters $\varepsilon, \delta > 0$ for the FPAUS. Let $m \in \text{poly}(n)$ be such that for any $y \in R(x)$ we have $|y| \leq m(|x|)$. By Lemma D.3 and the assumption of CoT model $\pi$, for every position $i \in [m(|x|)]$, every input $x \in \Sigma^*$, and every prefix $y_{<i}$, it holds that

$$\Pr_\omega \Big[ (1 - \tfrac{1}{2m(|x|)}) \, p(y_i \mid x, y_{<i}) \leq \pi_\omega(y_i \mid x, y_{<i}) \leq (1 + \tfrac{1}{m(|x|)}) \, p(y_i \mid x, y_{<i}) \Big] \geq 1 - \frac{\delta}{3m(|x|)}, \tag{76}$$

where $\pi_\omega$ denotes the distribution obtained by taking the median over multiple trials, and $\omega$ represents the underlying randomness of these trials.

Hence the autoregressive sampling of $y_1, \dots, y_{m(|x|)}$ succeeds at all steps with probability at least

$$1 - m(|x|) \cdot \tfrac{\delta}{3m(|x|)} = 1 - \tfrac{\delta}{3}, \tag{77}$$

by the union bound. Condition on this "good" event. Let $p(y \mid x) = 1/|R(x)|$ denote the uniform target on $R(x)$. Multiplying the stepwise bounds and using Lemma D.4, namely $\left(1 + \tfrac{1}{2m}\right)^m \leq e^{1/2}$, we obtain

$$e^{-1/2} \leq \frac{\pi_\omega(y \mid x)}{p(y \mid x)} \leq e^{1/2} \qquad \text{for all } y \in R(x). \tag{78}$$

To obtain exact uniformity, we apply rejection sampling. Run one preliminary draw $z \sim \pi_\omega(\cdot \mid x)$ and set $\varphi_0 := e^{-1} \pi_\omega(z \mid x)$, and define the acceptance probability, denoted by $a(y)$, as

$$a(y) := \min \Big\{ 1, \frac{\varphi_0}{\pi_\omega(y \mid x)} \Big\}. \tag{79}$$

By equation 78 applied to both $z$ and any $y \in R(x)$,

$$\pi_{\boldsymbol{\omega}}(z \mid x) \leq \mathrm{e}^{1/2}\, p(z \mid x) = \frac{\mathrm{e}^{1/2}}{|R(x)|} \quad \text{and} \quad \pi_{\boldsymbol{\omega}}(y \mid x) \geq \mathrm{e}^{-1/2}\, p(y \mid x) = \frac{\mathrm{e}^{-1/2}}{|R(x)|}. \quad (80)$$

Therefore

$$\varphi_0 = \mathrm{e}^{-1}\pi_{\boldsymbol{\omega}}(z \mid x) \leq \frac{\mathrm{e}^{-1/2}}{|R(x)|} \leq \pi_{\boldsymbol{\omega}}(y \mid x) \qquad \text{for all } y \in R(x). \quad (81)$$

Thus $a(y)$ is well-defined, and the output probabilities of the rejection sampling procedure satisfy

$$\Pr[y] \propto \pi_{\boldsymbol{\omega}}(y \mid x)\, a(y) = \min\{\pi_{\boldsymbol{\omega}}(y \mid x), \varphi_0\}. \quad (82)$$

which is constant in $y$. Hence, conditional on the good event, the output distribution is *exactly* uniform over $R(x)$. Still, on the good event, the acceptance probability in one trial is

$$\Pr[\text{accept}] = \sum_{y \in R(x)} \pi_{\boldsymbol{\omega}}(y \mid x)\, a(y) = |R(x)|\,\varphi_0 \geq |R(x)| \cdot \mathrm{e}^{-1} \cdot \frac{\mathrm{e}^{-1/2}}{|R(x)|} = \mathrm{e}^{-3/2}. \quad (83)$$

Hence, the per-trial rejection probability is at most $q := 1 - \mathrm{e}^{-3/2}$. Repeating independently $k$ times, the probability of rejecting in all trials is at most $q^k$. To make this at most $\delta/3$, it suffices to take

$$k \geq \log(3/\delta)/\log(1/q) = O(\log(1/\delta)). \quad (84)$$

There are three failure modes: (i) stepwise approximation fails (probability $\leq \delta/3$); (ii) the preliminary draw does not satisfy equation 78 (already covered by (i), so no extra cost); (iii) all $k$ rejection-sampling trials reject ($\leq \delta/3$). By the union bound, the overall failure probability is at most $\delta$. On success, the output law is uniform, so its total variation distance to the uniform target is also at most $\delta$. The running time is polynomial in $|x|$ and $\log(1/\delta)$, so the procedure is an FPAUS. $\qquad \square$

## D.3 PROOF FOR THEOREM 4.9

First, we show the expressiveness of CoT with stochastic decoding, which can simulate a probabilistic Turing machine (PTM). For simplicity, we consider a two-tape PTM.

**Definition D.7.** A probabilistic Turing machine $M = (Q, \Sigma, \Gamma, q_0, q_1, A, \delta_1, \delta_2)$ is defined as:

- $Q$ is a finite set of states,

- $\Sigma$ is the input/output alphabet,

- $\Gamma$ is the tape alphabet, with $\Sigma \subseteq \Gamma$ and a distinguished blank symbol $\sqcup \in \Gamma$,

- $q_0 \in Q$ denotes the initial state, and $q_1 \in Q$ denotes the final state,

- $\delta_1, \delta_2 : Q \times \Gamma \to Q \times \Gamma \times (\Sigma \cup \{\varepsilon\}) \times \{L, S, R\}$ are two transition functions.

At each step, $M$ chooses uniformly at random between $\delta_1$ and $\delta_2$. Each transition $\delta_i(q, a) = (q', b, \sigma, D)$ is interpreted as: writing $b \in \Gamma$ on the work tape, writing $\sigma \in \Sigma$ on the output tape, where $\varepsilon$ means that nothing is written, and moving the tape head in direction $D \in \{L, S, R\}$, where $L =$ left, $R =$ right, and $S =$ stay. The output of $M$ is defined to be the string remaining on the output tape when the machine halts.

Our result builds on Nowak et al. (2024), who extended that of Pérez et al. (2021) from simulating deterministic Turing machines with attention to incorporating probabilistic decoding, thereby capturing the probabilistic transitions of PTMs. Their proof, however, relies on hard attention, whereas our setting uses softmax attention. In addition, the definition of the output format of CoT models considered here differs from theirs. We therefore establish the corresponding statements in our setting.

**Lemma D.8.** *Let $M$ be a PTM with input and output alphabet $\Sigma$, and running time bounded by $T(n)$ on inputs of length $n$. Then there exists a log-precision CoT model with stochastic decoding, denoted $\mathsf{CoT}_M$, such that the induced output distribution of $\mathsf{CoT}_M$ coincides exactly with that of $M$. Formally, for every input string $x \in \Sigma^*$ and output string $y \in \Sigma^*$,*

$$\Pr\big[\mathsf{CoT}_M(x) = y\big] = \Pr\big[M(x) = y\big].$$

*Moreover, the number of reasoning steps of $\mathsf{CoT}_M$ is bounded by $O(T(|x|))$.*

*Proof.* We argue by induction on the number of machine steps and construct a CoT whose token sequence mirrors the PTM's execution. Let $Q$ be the state set of $M$, $\Gamma$ the work-tape alphabet, and $\Sigma$ the input/output alphabet. At each step $i \geq 0$, the machine has a configuration summarized by

$$\boldsymbol{r}_i = \left(q^{(i)}, v^{(i)}, w^{(i)}, m^{(i)}\right) \in Q \times \Gamma \times (\Sigma \cup \{\varepsilon\}) \times \{-1, 0, 1\},$$

where $q^{(i)}$ is the current state, $v^{(i)}$ is the symbol written on the work tape (at the head cell) during step $i$, $w^{(i)}$ is the symbol written to the output tape (or $\varepsilon$ if no output), and $m^{(i)} \in \{-1, 0, 1\}$ is the head move. Let the output string be $y = (y_1, \ldots, y_m) \in \Sigma^*$, and let $r : [m] \to \mathbb{N}$ map each output index $i$ to the index of step $r(i)$ in which $M$ writes $y_i$ to the output tape.

The precision of CoT is $s(n) = C \cdot \lceil \log_2 n \rceil$, where $C \in \mathbb{N}$ is chosen sufficiently large so that $n^C$ exceeds the maximum polynomial step bound under consideration. For input $x = (x_1, \ldots, x_n) \in \Sigma^n$, the model produces the sequence

$$\left(x, \ \boldsymbol{r}_{1:r(1)}, \ e, \ y_1, \ e', \ \boldsymbol{r}'_{r(1)}, \ \boldsymbol{r}_{r(1)+1:r(2)}, \ e, \ y_2, \ e', \ \boldsymbol{r}'_{r(2)}, \ \ldots, \ \boldsymbol{r}_{r(m-1)+1:r(m)}, \ e, \ y_m, \ e'\right), \tag{85}$$

where blocks of configuration tokens $\boldsymbol{r}_i$ simulate the PTM step-by-step and the triple $(e, y_i, e')$ deterministically realizes the $i$-th output when it occurs. The vocabulary of the model is

$$\mathcal{V} := Q \times \Gamma \times (\Sigma \cup \{\varepsilon\}) \times \{-1, 0, 1\} \times \{r, r'\} \ \cup \ (\Sigma \cup \{\varepsilon\}) \ \cup \ \{e, e'\}. \tag{86}$$

Here, $\{r, r'\}$ is a binary flag used to distinguish whether a configuration token carries the ordinary marker $r$ or the primed marker $r'$.

**Embeddings** Let $\boldsymbol{e}_Q : Q \to \{0, 1\}^{|Q|}$, $\boldsymbol{e}_\Gamma : \Gamma \to \{0, 1\}^{|\Gamma|}$, and $\boldsymbol{e}_\Sigma : \Sigma \to \{0, 1\}^{|\Sigma|}$ denote one-hot encodings, and let $\boldsymbol{p}(\cdot) = \mathrm{sbin}_{s(n)}(\cdot)$ be the positional code. We write $\boldsymbol{0}$ for a zero block of appropriate size. The embedding of an input token $x_i$ is

$$\left(\boldsymbol{e}_Q(q^{(0)}), \ \boldsymbol{e}_\Gamma(x_i), \ \boldsymbol{0}_{|\Sigma|}, \ 0, \ \boldsymbol{p}(i), \ \boldsymbol{0}\right). \tag{87}$$

For a configuration token $\boldsymbol{r}_i = (q^{(i)}, v^{(i)}, w^{(i)}, m^{(i)})$ at position $n + j$, the embedding is

$$\left(\boldsymbol{e}_Q(q^{(i)}), \ \boldsymbol{e}_\Gamma(v^{(i)}), \ \boldsymbol{e}_\Sigma(w^{(i)}), \ m^{(i)}, \ \boldsymbol{p}(j), \ \boldsymbol{0}\right). \tag{88}$$

For delimiters $d \in \{e, e'\}$ at position $n + j$, we use

$$\left(\boldsymbol{0}_{|Q|}, \ \boldsymbol{0}_{|\Gamma|}, \ \boldsymbol{0}_{|\Sigma|}, \ 0, \ \boldsymbol{p}(j), \ \boldsymbol{p}(j-1), \ \boldsymbol{p}(j-3) \ \boldsymbol{0}, \ \mathbf{1}[d = e], \ 0, \ \mathbf{1}[d = e'], \ \boldsymbol{0}_{|\mathcal{V}|}\right). \tag{89}$$

Finally, for an output token $y_i \in \Sigma$ at position $n + j$, the embedding is

$$\left(\boldsymbol{0}_{|Q|}, \ \boldsymbol{0}_{|\Gamma|}, \ \boldsymbol{e}_\Sigma(y_i), \ 0, \ \boldsymbol{p}(j), \ \boldsymbol{0}, \ 0, \ 1, \ 0, \ \boldsymbol{0}_{|\mathcal{V}|}\right). \tag{90}$$

**First layer** For each input position $i$, the model computes the head position of the Turing machine both at and immediately after the $i$-th step. This construction follows the approach of Pérez et al. (2021); Merrill & Sabharwal (2024); Nowak et al. (2024). The next head position is obtained by summing the head movement directions $m^{(j)} \in \{-1, 0, 1\}$ up to step $i$:

$$c^{(i+1)} = m^{(0)} + m^{(1)} + \cdots + m^{(i)}. \tag{91}$$

To realize this with single-head attention, we define the query, key, and value vectors at position $j$ as

$$\mathbf{q}_j = (1), \quad \mathbf{k}_j = (1), \quad \mathbf{v}_j = (m^{(j)}). \tag{92}$$

Since the attention scores are uniform across all positions, the attention layer computes the cumulative sum of head movements. A subsequent feed-forward layer then yields $c^{(i)} = c^{(i+1)} - m^{(i)}$. Thus, at token $\boldsymbol{r}_k$ at position $n + i$, the representation is

$$\left(\boldsymbol{e}(q^{(k)}), \ \boldsymbol{e}(v^{(k)}), \ \boldsymbol{e}(w^{(k)}), \ m^{(k)}, \ \boldsymbol{p}(i), \ c^{(k)} \ \boldsymbol{0}\right). \tag{93}$$

**Second layer** For configuration $r_k$, the model retrieves the symbol located under the head, denoted by $s^{(k)}$. To accomplish this, we define an attention mechanism that filters positions $j \leq k$ such that $c^{(j)} = c^{(k)}$, and selects the largest such index $j$. Formally, $s^{(k)} = v^{(j^\star)}$, where $j^\star = \max\{j \leq k : c^{(j)} = c^{(k)}\}$. To implement this retrieval, we set the attention vectors at position $j$ as

$$\mathbf{q}_j = \Big(\mathsf{sbin}_{s(n)}(j) ^\frown 1_{s(n)}, \ T(n) \cdot j\Big), \tag{94}$$

$$\mathbf{k}_j = 2^{s(n)+1} \cdot \Big(\mathsf{sbin}_{s(n)}(j)^\frown -1_{s(n)}, \ T(n) \cdot j\Big), \quad \mathbf{v}_j = \Big(e_\Gamma(v^{(j)}), \ \mathbf{0}\Big). \tag{95}$$

Here, the construction guarantees that there are only $T(n) \cdot O(\log n)$ distinct scores. Although the resulting attention is not a perfect one-hot vector, the ordering of scores is preserved, so the maximum weight still corresponds to the correct index. Consequently, the model can reliably retrieve a value that approximates $e_\Gamma(s^{(k)})$. The subsequent feed-forward layer computes the two transition functions $\delta_1$ and $\delta_2$ in parallel:

$$z_j^{(k+1)} = e\big(\delta_j\big(q^{(k)}, s^{(k)}\big)\big) := \big(e_Q(q_j^{(k+1)}), e_\Gamma(v_j^{(k+1)}), e_\Sigma(w_j^{(k+1)}), m_j^{(k+1)}\big), \quad j \in \{1, 2\}. \tag{96}$$

Finally, the overall output at token $r_k$ at position $n + i$ is

$$\Big(e(q^{(k)}), \ e(v^{(k)}), \ e(w^{(k)}), \ m^{(k)}, \ p(i), \ c^{(k)}, \ s^{(k)}, \ z_1^{(k+1)}, \ z_2^{(k+1)}, \ \mathbf{0}\Big), \tag{97}$$

where $z_1^{(k+1)}$ and $z_2^{(k+1)}$ represent the encodings of the two candidate transitions.

**Third layer** This layer handles the delimiters. When the last input token is $e$, the model copies the most recently written output symbol $y_i$ from the immediately preceding position. When the last input token is $e'$, the model copies the most recent configuration $\mathsf{v}_i$ from three positions earlier. We implement this mechanism using two attention heads:

$$\mathbf{q}_{i,1} = \mathsf{sbin}_{s(n)}(i-1) ^\frown 1_{s(n)}, \tag{98}$$

$$\mathbf{q}_{i,2} = \mathsf{sbin}_{s(n)}(i-3) ^\frown 1_{s(n)}, \tag{99}$$

$$\mathbf{k}_{i,1} = \mathbf{k}_{i,2} = 2^{s(n)+1} \cdot \mathsf{sbin}_{s(n)}(i) ^\frown (-1_{s(n)}), \tag{100}$$

$$\mathbf{v}_{i,1} = \mathbf{v}_{i,2} = \big(e(q^{(i)}), e(v^{(i)}), e(w^{(i)}), m^{(i)}, \mathbf{0}\big). \tag{101}$$

The first head is configured to retrieve the symbol from one position earlier, while the second head retrieves the configuration from three positions earlier. The following feed-forward map, conditioned on the one-hot flags indicating whether the input token is one of $\{r_i, e, y_i, e'\}$, outputs $\big(*, \ z\big)$, where $z \in \{0, 1\}^{|\mathcal{V}|}$ is defined casewise as

$$z = \begin{cases} \frac{1}{2}\Big(e\big(z_1^{(k+1)}\big) + e\big(z_2^{(k+1)}\big), \ \mathbf{0}_{|\Sigma|+4}\Big), & \text{if the input is } r_i \text{ and } i \neq r(t) \ \forall t, \\ \Big(\mathbf{0}_{|Q| \times |\Gamma| \times (|\Sigma|+1) \times 6}, \ \mathbf{0}_{|\Sigma|+1}, \ (1, 0, 0)\Big), & \text{if the input is } r_i \text{ and } i = r(t) \text{ for some } t, \\ \Big(\mathbf{0}_{|Q| \times |\Gamma| \times (|\Sigma|+1) \times 6}, \ e_\Sigma(y_i), \ \mathbf{0}_3\Big), & \text{if the input is } e, \\ \Big(\mathbf{0}_{|Q| \times |\Gamma| \times (|\Sigma|+1) \times 6}, \ \mathbf{0}_{|\Sigma|+1}, \ (0, 0, 1)\Big), & \text{if the input is } y_i, \\ \Big(e(r'_{r(i)}), \ \mathbf{0}_{|\Sigma|+4}\Big), & \text{if the input is } e'. \end{cases} \tag{102}$$

Here $e(\cdot)$ denotes one-hot encodings aligned with the corresponding slots of $\mathcal{V}$.

**Output layer** The final output layer is $\mathbf{OUT} = \begin{bmatrix} \mathbf{0} & \boldsymbol{I}_{|\mathcal{V}|} \end{bmatrix}$. The decoding function generates tokens in two modes: when the current token is $r_i$, it stochastically samples from the two transitions $z_1^{(k+1)}$ and $z_2^{(k+1)}$; otherwise, it deterministically outputs $y_i$ for $e$, $e'$ for $y_i$, and $r_{r(i)}$ for $e'$.

$\square$

*Proof for Theorem 4.9.* Under the assumption FPTAS $\subsetneq$ FPRAS for self-reducible relations, let $R$ denote a self-reducible relation whose counting problem $|R(x)|$ admits an FPRAS but not an FPTAS.

**Proof of the CoT Part**  Define the extension counting function

$$\mathrm{EXT}_R : \Sigma^* \times \Sigma^* \to \mathbb{N}, \qquad \mathrm{EXT}_R(x, w) \;=\; \big|\{\, z \in \Sigma^* : \langle x, wz \rangle \in R \,\}\big|.$$

First, we briefly show that if there exists an FPRAS for $|R(x)|$, then there also exists an FPRAS for $\mathrm{EXT}_R$, as shown by (Jerrum et al., 1986). Let $\mathcal{A}(x, \varepsilon, \delta)$ denote an FPRAS for the counting problem $|R(x)|$, i.e., for any $\varepsilon, \delta \in (0, 1)$,

$$\Pr[(1 - \varepsilon)\,|R(x)| \;\leq\; \mathcal{A}(x, \varepsilon, \delta) \;\leq\; (1 + \varepsilon)\,|R(x)|] \;\geq\; 1 - \delta. \tag{103}$$

By self-reducibility, given $(x, w)$ we proceed as follows: while $|w| > \sigma(x)$, decompose $w = uv$ with $|u| = \sigma(x)$, set $x \leftarrow \psi(x, u)$ and $w \leftarrow v$. When the loop terminates, define $S := \Sigma^{\sigma(x) - |w|}$. Then

$$\mathrm{EXT}_R(x, w) \;=\; \sum_{u \in S} |R(\psi(x, wu))|.$$

An $(\varepsilon, \delta)$-relative estimate of $\mathrm{EXT}_R(x, w)$ is obtained by replacing each summand with $\mathcal{A}(\psi(x, wu), \varepsilon, \delta/|S|)$ and summing. A union bound guarantees overall relative error within $(1 \pm \varepsilon)$ with probability at least $1 - \delta$. Let $\widehat{\mathrm{EXT}}_R(\cdot, \cdot)$ be these $(1 \pm \varepsilon)$ estimates, and define

$$\pi(y_i \mid x, y_{<i}) \;:=\; \frac{\widehat{\mathrm{EXT}}_R(x, y_{<i} y_i)}{\sum_{u \in \Sigma} \widehat{\mathrm{EXT}}_R(x, y_{<i} u)} \qquad (y_i \in \Sigma). \tag{104}$$

The target (uniform) conditional distribution is

$$p(y_i \mid x, y_{<i}) \;:=\; \frac{\mathrm{EXT}_R(x, y_{<i} y_i)}{\sum_{u \in \Sigma} \mathrm{EXT}_R(x, y_{<i} u)} \qquad (y_i \in \Sigma). \tag{105}$$

On the event that all estimates in equation 104 are $(1 \pm \varepsilon)$-accurate, we obtain

$$\frac{1 - \varepsilon}{1 + \varepsilon}\, p(y_i \mid x, y_{<i}) \;\leq\; \pi(y_i \mid x, y_{<i}) \;\leq\; \frac{1 + \varepsilon}{1 - \varepsilon}\, p(y_i \mid x, y_{<i}) \qquad \text{for all } y_i \in \Sigma. \tag{106}$$

Choosing $\varepsilon \leq \frac{1}{2\alpha(|x|)+1}$ for any polynomial $\alpha(n) \in \mathsf{poly}(n)$ yields

$$\frac{1 + \varepsilon}{1 - \varepsilon} \leq 1 + \frac{1}{\alpha(|x|)}, \qquad \frac{1 - \varepsilon}{1 + \varepsilon} \geq 1 - \frac{1}{\alpha(|x|)},$$

thus, we have an approximation to the true conditional distribution.

Finally, let $m(n) \in \mathsf{poly}(n)$ be such that $|y| \leq m(|x|)$ for all $y \in R(x)$. Applying a union bound over the $m(n)$ calls in Equation (104), if each call uses failure probability $\delta \leq \frac{\frac{1}{2} - \gamma}{m(|x|)}$ for any fixed $\gamma \in (0, \frac{1}{2})$, then the joint success event holds with probability at least $1 - (|\Sigma| + 1)\delta \geq \frac{1}{2} + \gamma$. Thus there exists an FPRAS ensuring

$$\Pr\left[(1 - \tfrac{1}{\alpha(|x|)}) p(y_i \mid x, y_{<i}) \;\leq\; \pi(y_i \mid x, y_{<i}) \;\leq\; (1 + \tfrac{1}{\alpha(|x|)}) p(y_i \mid x, y_{<i})\right] \;\geq\; \tfrac{1}{2} + \gamma. \tag{107}$$

By Lemma D.8, any PTM that realizes an FPRAS can be simulated by a log-precision CoT model with stochastic decoding, whose output distribution coincides exactly with that of the PTM. Hence, there exists a CoT model with polynomially many reasoning steps that achieves the $(\alpha, \gamma)$-weak probable approximation to the uniform distribution $U(R(x))$.

**Proof of the looped TF Part**  Suppose, for contradiction, that there exists a looped TF $\pi$ with polynomial loops, which, for every input $x$, achieves an $(\alpha, \gamma)$-weak probable approximation to the uniform distribution $U(R(x))$ with

$$\alpha(|x|) \;\geq\; \frac{\max\{\log(1 + \varepsilon),\; \log(1 - \varepsilon) + \frac{1}{2}\}}{m(|x|)}, \tag{108}$$

where $m(n) \in \mathsf{poly}(n)$ satisfies $|y| \leq m(|x|)$ for any $y \in R(x)$, with $\varepsilon > 0$ and constant $\gamma < 1/2$.

By definition of the model, all randomness is injected only during decoding. The logits, and thus the conditional distributions $\pi(\cdot \mid x, y_{<i})$, are deterministically computable in time $\text{poly}(|x|)$. Hence, the guarantee reduces to a deterministic bound: for every $i$ and prefix $y_{<i}$,

$$(1 - 1/\alpha(|x|))\, p(y_i \mid x, y_{<i}) \;\leq\; \pi(y_i \mid x, y_{<i}) \;\leq\; (1 + 1/\alpha(|x|))\, p(y_i \mid x, y_{<i}). \qquad (109)$$

For a fixed string $\hat{y} \in R(x)$, define the polynomial-time computable quantity

$$\widehat{Z}(x) \;:=\; \left( \prod_{i=1}^{|\hat{y}|} \pi(\hat{y}_i \mid x, \hat{y}_{<i}) \right)^{-1}. \qquad (110)$$

Since $p(y \mid x) = 1/|R(x)| = \prod_i p(y_i \mid x, y_{<i})$, we have

$$(1 - 1/\alpha(|x|))^m \;\leq\; \frac{\widehat{Z}(x)}{|R(x)|} \;\leq\; (1 + 1/\alpha(|x|))^m. \qquad (111)$$

By Lemma D.4, whenever $\varepsilon^2 \leq m$,

$$e^{\varepsilon - \frac{1}{2}} \;\leq\; \left( 1 + \tfrac{\varepsilon}{m} \right)^m \;\leq\; e^{\varepsilon}. \qquad (112)$$

Substituting $\varepsilon = \max\{\log(1 + \varepsilon),\, \log(1 - \varepsilon) + \tfrac{1}{2}\}$ into the inequality, we have

$$(1 - \varepsilon)\, |R(x)| \;\leq\; \widehat{Z}(x) \;\leq\; (1 + \varepsilon)\, |R(x)|, \qquad (113)$$

i.e., a deterministic $(1 \pm \varepsilon)$-approximation to $|R(x)|$ computable in time $\text{poly}(|x|, 1/\varepsilon)$. This constitutes an FPTAS, contradicting the assumption that FPTAS $\subsetneq$ FPRAS.

From the preceding argument, there exist a polynomial function $\alpha(n) \in \text{poly}(n)$ and a constant $\gamma > 0$ such that, for any input $x$, any looped TF with polynomial loops has a first-step conditional $\pi(\cdot \mid x)$ that deviates from the target $p(\cdot \mid x) = U(R(x))$ on a set of symbols of non-negligible mass. Formally, define

$$A \;:=\; \left\{ a \in \Sigma \;\;\middle|\;\; \frac{\pi(a \mid x)}{p(a \mid x)} \leq 1 - \frac{1}{\alpha(|x|)} \;\; \text{or} \;\; \frac{\pi(a \mid x)}{p(a \mid x)} \geq 1 + \frac{1}{\alpha(|x|)} \right\}, \qquad (114)$$

and assume $p(A \mid x) \geq \gamma$. Let $q(y \mid x) := \prod_{k=1}^m \pi(y_k \mid x, y_{<k})$ be the model's full distribution over strings. By data processing (marginalization cannot increase total variation),

$$\|q(\cdot \mid x) - p(\cdot \mid x)\|_{\text{TV}} \;\geq\; \|\pi(\cdot \mid x) - p(\cdot \mid x)\|_{\text{TV}} \;=\; \frac{1}{2} \sum_{a \in \Sigma} |\pi(a \mid x) - p(a \mid x)|. \qquad (115)$$

For every $a \in A$ we have $|\pi(a \mid x) - p(a \mid x)| = p(a \mid x)\left|\frac{\pi(a|x)}{p(a|x)} - 1\right| \geq \frac{1}{\alpha(|x|)}\, p(a \mid x)$. Hence

$$\|q(\cdot \mid x) - p(\cdot \mid x)\|_{\text{TV}} \;\geq\; \frac{1}{2} \sum_{a \in A} \frac{1}{\alpha(|x|)}\, p(a \mid x) \;\geq\; \frac{\gamma}{2\, \alpha(|x|)}. \qquad (116)$$

Therefore, the total variation distance from $U(R(x))$ is bounded below by $\gamma/(2\alpha(|x|))$ and thus cannot be made arbitrarily small. In particular, such a model cannot realize an FPAUS for $U(R(x))$. $\qquad \square$

# E  EXPERIMENTAL DETAILS

## E.1  PARALLELIZABLE TASKS

### E.1.1  TASK SETTINGS

**Word Problem**  We define a sequence prediction task based on finite groups such as the symmetric group $S_5$. Given a sequence of group elements of length $k$, the model is required to output the cumulative products obtained by scanning the sequence from left to right. Formally, for an input sequence $(g_1, g_2, \ldots, g_k)$, the target sequence is $(g_1,\; g_1 g_2,\; g_1 g_2 g_3,\; \ldots,\; g_1 g_2 \cdots g_k)$. We follow the setting of Merrill & Sabharwal (2025b).

**Connectivity** To ensure the dataset has balanced labels, we generate undirected graphs according to the Erdős–Rényi model (Erdos & Renyi, 1959) $G(n, p)$, where $n$ is the number of vertices and each possible edge is included independently with probability $p$. In the supercritical regime ($pn = c > 1$), a single "giant" connected component emerges, occupying a fraction $s \in (0, 1)$ of the vertices, which satisfies $s = 1 - e^{-cs}$. Consequently, the probability that two uniformly random vertices are both in this component—and hence mutually reachable—is approximately $s^2$.

To target a reachability probability of $1/2$, we set $s \approx \sqrt{\frac{1}{2}} \approx 0.707$, $\quad c \approx \frac{-\ln(1-s)}{s} \approx 1.74$, and thus $p = \frac{c}{n} \approx \frac{1.7}{n}$. In practice, for each graph of size $n$ we fix $p = 1.7/n$, which empirically yields $\Pr[\text{reachable}] \approx 50\%$ for $n \in [50, 100]$, up to small fluctuations. Other random models (e.g., stochastic block models) can also be tuned to achieve such a balance, but the Erdős–Rényi choice offers the simplest parametrization. We follow the encoding scheme of Sanford et al. (2024a). The input to the model is serialized as a flat token sequence consisting of three parts:

$$\underbrace{v_0 \, v_1 \, \cdots \, v_{n-1}}_{\text{vertices}} \, \underbrace{e_1 \, e_2 \, \cdots \, e_m}_{\text{edges}} \, \underbrace{s, t}_{\text{query}},$$

where each vertex is denoted by a token $v_i$, each edge is represented as a pair "`u,v`" with $u < v$, and the final token "`s,t`" specifies the source–target pair for the reachability query.

**Arithmetic Expression Evaluation** Following Feng et al. (2023), we generate expressions over integers modulo $r$ using the four operations $+, -, \times, \div$, where multiplication and division are defined via precomputed modular tables. To guarantee that each expression evaluates to a specific target value, we grow expressions *backwards*: starting from a sampled number, we iteratively replace it with a binary sub-expression that preserves its value under modular arithmetic. Different from Feng et al. (2023), we fix the modulus to $r = 3$, as our focus lies in evaluating the reasoning over expressions rather than exploring the properties of each modular arithmetic system.

**Edit Distance** The Edit Distance task requires computing the minimum number of edit operations needed to transform one string into another. The allowed operations are insertion, deletion, and replacement of a single character, and the objective is to predict the total edit distance given two input strings. To build the dataset, we follow Feng et al. (2023). We first generate two strings over a randomly sampled alphabet. The first string has a fixed length, while the second string is produced in two possible ways: with probability $0.4$, it is drawn as a random string of nearly the same length (within $\pm 3$ characters), and with probability $0.6$, it is derived from the first string by applying a given number of random edit operations. Each edit operation is chosen uniformly among deletion, replacement, and insertion. To avoid trivial cases, string pairs that are identical or whose lengths differ excessively are rejected and resampled.

### E.1.2 Sequential Algorithms for CoT

**(1) Word problem:** for this task, the CoT algorithm proceeds by sequentially scanning the token sequence and producing at each prefix the evaluation result of the expression step by step. Thus, the overall length of the CoT sequence matches the input length. **(2) Graph connectivity:** Following Bavandpour et al. (2025), the algorithm sequence is simply the trace of a breadth-first search (BFS) starting from the source $s$. At each step, the model emits the incident edges of the currently expanded node in the order they are visited. The sequence terminates as soon as the target $t$ is discovered. To implement this algorithm, we maintain a list ("scratchpad") initialized with a dummy marker and the source, $(\text{N}, s)$. We iterate through this list from left to right (i.e., queue order). Whenever the current node $u$ is expanded, we append to the end of the list all incident edges $(u, v)$ for neighbors $v$, optionally followed by a separator token $(u, \text{N})$. **(3) Arithmetic expression evaluation:** Following Feng et al. (2023), the CoT takes the fully expanded expression and repeatedly evaluates one innermost subexpression, writing down the simplified expression at each step until only a single numeral remains. For example, $2 * (0 + 1)/2 \to 2 * 1/2 \to 2/2 \to 1$. The overall CoT sequence has quadratic length. **(4) Edit distance:** Following Feng et al. (2023), the CoT algorithm outputs the DP table entries in the same order they are computed, i.e., row by row from top-left to bottom-right (topological order). This yields a quadratic-length CoT sequence.

### E.1.3 TRAINING CONFIGURATION

We trained all models using the AdamW optimizer with a linear learning rate schedule. The initial learning rate was set to $1 \times 10^{-4}$ with a weight decay of $0.01$, and a batch size of $256$. Training was continued until the training loss plateaued. For looped TFs, curriculum learning was applied to all tasks except edit distance: the input size was increased by $2$ for the word problem task, and by $4$ for the connectivity and arithmetic evaluation tasks. The model architecture was based on standard Transformers with an embedding dimension of $256$. We used $4$ attention heads, and varied the number of Transformer layers depending on the task: two layers for word problems, a single layer for connectivity, and time-modulated (Xu & Sato, 2025) model with a single layer for looped TF on both arithmetic evaluation and the edit distance task. For CoT, we use the same configuration of the Transformer block.

### E.2 APPROXIMATION TASK

To generate the dataset, we first construct a DNF formula $F$ by sampling $m$ clauses, each consisting of $w$ distinct literals over $n$ Boolean variables. Each literal is independently assigned to be either positive or negated. The formula is then serialized into a token sequence in which each clause is represented by its index together with variable–value pairs such as "$2 = +1$" or "$4 = -1$". For looped TFs, we prepare $100{,}000$ training samples and $1{,}000$ test samples. For CoT models, we instead generate an online dataset with the following structure. To train the CoT models, we simulate a single trial of the randomized counting algorithm of Karp & Luby (1983) as follows:

1. A clause $C_1$ is sampled with probability proportional to $2^{-|C_j|}$ (uniform in our experiments).

2. An assignment consistent with $C_1$ is generated by fixing the variables in the clause $C_1$ and randomly setting the remaining variables.

3. A clause $C_2$ is then selected uniformly at random, and the assignment is checked against it.

4. If the assignment satisfies $C_2$, the outcome token is set to Success; otherwise, Fail.

The sequence concatenates the serialized formula, the sampled clause, the full assignment, and the verification outcome, separated by <sep> tokens and terminated by <eos>. The CoT model is trained in an autoregressive manner, where prediction targets are defined by shifting the token sequence while masking out the formula description. We trained the models using the AdamW optimizer with a linear learning rate schedule. The initial learning rate was set to $1 \times 10^{-4}$, with a weight decay of $0.01$. We used a batch size of $256$ (reduced to $32$ for the 1000-loop setting) and trained for $10{,}000$ iterations.

## F ON THE FAIRNESS OF COMPARING LOOP COUNT AND COT STEPS

CoT requires far less computation by reusing the KV cache; however, this optimization shifts the bottleneck to memory bandwidth, leaving computational units under-utilized. In contrast, a looped TF recomputes all positions at each loop, incurring higher computational cost, but its computation is compute-bound and can fully utilize GPU parallelism. Thus, while the computational costs differ, the comparison is justified from a time-to-solution perspective, as the effective wall-clock throughput of the two approaches can be comparable under realistic hardware constraints.

