# OpenReview forum: "A Formal Comparison Between Chain-of-Thought and Latent Thought"
_ICLR.cc/2026/Conference — Submitted to ICLR 2026_

### Official Review · Reviewer_hvpQ · 2025-10-30

**Soundness:** 3
**Presentation:** 2
**Contribution:** 2
**Rating:** 2
**Confidence:** 4

**Summary:**

This paper compares two reasoning paradigms in large language models — Chain-of-Thought (CoT) and Latent Thought implemented via Looped Transformers. It provides a formal theoretical analysis showing that Looped Transformers can perform efficient parallel computation by operating in latent space, while CoT reasoning, being sequential, excels in stochastic or approximate problem solving.

**Strengths:**

he paper makes a valuable theoretical contribution by precisely linking CoT and Looped Transformers to formal complexity classes, providing a solid mathematical foundation to compare reasoning in latent versus linguistic space.

**Weaknesses:**

The main limitation lies in lack of conceptual clarity and practical interpretability. The theoretical analysis is rigorous, the motivation and intuition for non-specialist readers however are weak — the connection between abstract complexity classes (e.g., TCk, ACk) and practical model behaviors remains opaque. The formalism is heavy and often repeats known results under new notation rather than offering fresh insights.

Most importantly!!!!1 the experimental validation is **very very narrow and superficial**, relying on small-scale, synthetic tasks that do not convincingly demonstrate real-world implications. The comparison between “loops” and “steps” lacks fairness in terms of computational cost and hardware efficiency, which undermines the practical value of the claimed separation. The assumptions about “log-precision” and “polynomial embedding size” are idealized and may not hold in realistic settings.

Last,
while the paper claims to reveal “complementary strengths,” it fails to discuss how these insights can inform model design or prompt engineering, leaving the contribution mostly theoretical and of limited immediate impact.

**Questions:**

see above

---

> ### Author Response · Authors · 2025-11-20
> **Response by Author**
>
> > W2-1: Most importantly!!!!1 the experimental validation is very very narrow and superficial, relying on small-scale, synthetic tasks that do not convincingly demonstrate real-world implications.
>
> Since our primary contribution is theoretical, the objective of experiments is to validate our formal analysis. Such validation must rely on tasks whose computational complexity is precisely characterized, and such tasks are necessarily synthetic. This is a common evaluation methodology in previous work on formal analysis [1,2,4,5]. Moreover, several influential studies in this line of work include minimal or even no empirical experiments [3,6,7], reflecting the nature of the field rather than the limitations specific to our approach.
>
> > W2-2: The assumptions about “log-precision” and “polynomial embedding size” are idealized and may not hold in realistic settings.
>
> The assumptions are not only standard but in fact necessary for formal analyses of Transformer expressivity. They are widely adopted in prior work rather and not newly introduced in our study. The polynomial-size assumption [2,4,6,7] is inherited directly from circuit complexity, where classes such as NC and TC are defined over polynomial-size circuit families. The log-precision assumption [1-7] is needed for representing positional information. From a complexity-theoretic perspective, it is essential to analyze worst-case hardness, which is why these assumptions remain necessary in our proofs. However, our empirical observations indicate that our results still appear even when the assumptions are not strictly enforced.
>
> > W2-3: The comparison between “loops” and “steps” lacks fairness in terms of computational cost and hardware efficiency, which undermines the practical value of the claimed separation.
>
> From a hardware-constrained perspective, the comparison is justified in terms of _time_ rather than _computational cost_. Although CoT reduces FLOPs by reusing the KV-cache, this optimization makes inference heavily memory-bound: much of the GPU’s compute capacity remains idle, and throughput becomes limited by memory bandwidth rather than arithmetic operations. In contrast, a Looped Transformer can actively utilize otherwise idle compute resources, exploiting GPU parallelism far more effectively. As a result, its higher computational cost can still translate into comparable throughput to CoT. We will clarify this point in the appendix.
>
> > W3: Last, while the paper claims to reveal “complementary strengths,” it fails to discuss how these insights can inform model design or prompt engineering, leaving the contribution mostly theoretical and of limited immediate impact.
>
> We acknowledge that “distinct strengths” is a more accurate characterization than “complementary strengths,” and we have revised the paper accordingly.
>
> ---
> [1] Feng et al., Towards Revealing the Mystery behind Chain of Thought (NeurIPS 2023)
> [2] Li et al., Chain of Thought Empowers Transformers to Solve Inherently Serial Problems (ICLR 2024)
> [3] Merrill et al., The Expressive Power of Transformers with Chain of Thought (ICLR 2024)
> [4] Sanford et al.,Transformers, parallel computation, and logarithmic depth (ICML 2024 Spotlight)
> [5] Merrill et al., A Little Depth Goes a Long Way (NeurIPS 2025)
> [6] Merrill et al., Exact Expressive Power of Transformers with Padding (NeurIPS 2025)
> [7] On the Reasoning Abilities of Masked Diffusion Language Models (ICLR 2026 under review)

---

### Official Review · Reviewer_aNPY · 2025-10-30

**Soundness:** 1
**Presentation:** 1
**Contribution:** 1
**Rating:** 2
**Confidence:** 3

**Summary:**

This paper investigates the concept of latent thought and attempts to provide a formal analysis demonstrating that latent thought emerges in Looped Transformers. It seeks to explore the distinction between Chain-of-Thought (CoT) reasoning and latent thought within this framework. The authors approach the problem by formalizing deterministic computations as graph evaluations and conclude that latent thought facilitates efficient parallel computation, whereas CoT supports randomized approximation.

Overall, the paper suffers from unclear writing and lacks a well-defined objective and motivation. The theoretical analysis is not logically structured, making it difficult to follow. The experimental design is also weak, providing unconvincing results and offering limited insights.

**Strengths:**

1. The explored problem of investigating the gap between Chain-of-Thought (CoT) and latent thought is somewhat useful for developing effective methods of enhancing latent CoT reasoning.

**Weaknesses:**

1. The paper writing is unclear, lacking a well-defined objective and motivation.
2. The theoretical analysis is not logically structured, making it difficult to follow.
3. The experimental design is weak, providing unconvincing results and offering limited insights.

**Questions:**

1. Beyond the Looped Transformer architecture, can the proposed theoretical analysis be generalized to other latent CoT structures?
2. The theoretical analysis is poorly presented. What are the connections among the proposed theorems and lemmas?
3. How does the experimental design answer the proposed research question in the introduction?

---

> ### Author Response · Authors · 2025-11-23
> **Response by Author**
>
> > W1: The paper writing is unclear, lacking a well-defined objective and motivation.
>
> As stated in the introduction, our goal is to understand how Chain-of-Thought (CoT) and Latent Thought (Looped Transformers) differ in expressive power, and whether one fundamentally outperforms the other.
> To clarify the motivation behind this comparison, we have revised the introduction to better present why these two paradigms should be compared and what precise research questions our work seeks to answer.
>
> > **updated (44-45) :**
> These reasoning approaches share the common idea of iteratively applying Transformers to enhance expressive power, which naturally raises a fundamental question:
> *What is the separation between Chain-of-Thought and Latent Thought (Looped Transformer)?*
>
> > **updated (52-53) :** While these results suggest that looped models can be more expressive than CoT, several fundamental questions remain:
> *Does this separation extend beyond the logarithmic regime?*
> *Are Looped Transformers always more expressive than CoT?*
>
> ---
>
> > W2: The theoretical analysis is not logically structured, making it difficult to follow.
> > Q2: The theoretical analysis is poorly presented. What are the connections among the proposed theorems and lemmas?
>
> We appreciate the reviewer’s comment. We agree that the connections between the theoretical results and the main claims were not presented clearly enough. To address this, we have revised to explicitly summarize how each result contributes to the main separation results and to clarify the logical structure of our analysis.
>
> > **updated (61-70) :**  For many reasoning tasks, not only the correctness of a solution but also the efficiency of computation is crucial. By formalizing general reasoning problems as DAG evaluation, we show that while CoT can solve these problems using a small number of parameters, Latent Thought (looped models) can solve them more efficiently (Sec.3.1).
> To rigorously establish the separation, we extend our analysis to the computational complexity-theoretic setting (Sec.3.2).
> We show that, given sufficient model size, looped models with $\log^k{n}$ loops achieve exactly the power of $\mathsf{TC}^k$, a canonical parallel computation class. By contrast, CoT with $\log^k{n}$ steps falls short of $\mathsf{TC}^k$, leading to a strict separation in computational power favoring Latent Thought).
> In summary, Latent Thought gains efficiency by exploiting parallelism, whereas CoT remains fundamentally sequential.
>
> ---
>
> > W3: The experimental design is weak, providing unconvincing results and offering limited insights.
> > Q3: How does the experimental design answer the proposed research question in the introduction?
>
> We would like to respectfully point out that the experiments are designed to validate the formal analysis. Specifically:
> 1. The results for parallelizable tasks show that Looped Transformers can solve these tasks with substantially fewer loops than the number of CoT steps required, empirically supporting our theoretical separation in favor of Looped Transformers beyond the logarithmic regime. This directly answers yes to the question:
> *Does this separation (favoring Looped models) extend beyond the logarithmic regime?*
> 2. Additionally, the DNF counting experiments confirm the advantage of stochastic CoT over deterministic Looped Transformers, consistent with the separation established in Section 4, answering no to the question:
> *Are Looped Transformers always more expressive than CoT?*
>
> To address W3 and Q3, we have revised the introduction of the Experiments section to clearly articulate how the experimental setup validates our theoretical results and directly answers the research questions posed in the paper.
>
> > **updated (380-385) :** This section provides empirical validation of our theoretical results. First, we evaluate parallelizable reasoning tasks, where our results show that Looped TFs can solve these problems with substantially fewer iterative operations than CoT, consistent with the separation established in Section 3. Second, we consider a computationally hard approximate counting task, where the advantage shifts: CoT’s stochastic sampling enables accurate approximation, whereas deterministic Looped TFs fail to match its performance, supporting the theoretical results in Section 4.
>
> ---
>
> > Q1: Beyond the Looped Transformer architecture, can the proposed theoretical analysis be generalized to other latent CoT structures?
>
> The key requirement for our results is the presence of iterative latent-state updates—that is, the model must refine its hidden representations without emitting explicit tokens at each step. This property is shared by a broader class of latent CoT or continuous reasoning architectures, such as continuous chain-of-thought models.

---

> ### Comment · Reviewer_aNPY · 2025-11-26
>
> Thank you for your responses and for revising the manuscript. I appreciate the clarifications and improvements, and I have raised the soundness and presentation scores accordingly. However, I don't believe the overall quality of the paper yet meets the publication standard. Therefore, I keep the overall score unchanged.

---

### Official Review · Reviewer_nkKB · 2025-10-31

**Soundness:** 3
**Presentation:** 3
**Contribution:** 2
**Rating:** 4
**Confidence:** 3

**Summary:**

This paper presents a formal comparison between Chain-of-Thought (CoT) reasoning and Latent Thought in Looped Transformers (Looped TFs). Through theoretical analysis and controlled experiments, the authors demonstrate that Looped TFs enable efficient parallel computation in deterministic settings, whereas CoT excels in stochastic approximation tasks. The work establishes asymptotic separations between the two paradigms, suggesting when each reasoning strategy is more suitable.

**Strengths:**

- **Clear theoretical framing:** The paper provides a rigorous and formal comparison between CoT and Looped TFs, filling an underexplored gap in reasoning analysis.
- **Insightful comparative analysis:** The asymptotic and empirical results jointly clarify when each reasoning paradigm is advantageous.
- **Strong motivation:** The study addresses an important and timely question in understanding reasoning mechanisms in LLMs.

**Weaknesses:**

- **Simplifying assumptions:** The theoretical setting (deterministic vs. stochastic) abstracts away several practical factors that may affect real model behavior.
- **Clarity issues:** Some formal sections (e.g., proof sketches and separation definitions) could use more intuitive explanation or visualization.

**Questions:**

- **Assumption Robustness:**  The theoretical analysis relies on several strong assumptions (e.g., polynomial-size graphs, log-precision). How sensitive are the conclusions to these assumptions? Would relaxing them change the separation results?
- **Empirical Validation:**  The experiments (e.g., DAG and DNF counting) are highly synthetic. Can the authors demonstrate results on realistic reasoning benchmarks to confirm the claimed separation holds empirically?

---

> ### Author Response · Authors · 2025-11-20
> **Response by Author**
>
> > W1: Simplifying assumptions: The theoretical setting (deterministic vs. stochastic) abstracts away several practical factors that may affect real model behavior.
>
> We would like to emphasize that our contribution is to reveal the strength of CoT that were overlooked in prior work, which can only be uncovered by analyzing the stochastic setting. Previous work only compared deterministic Looped models with deterministic CoT, whereas our study is the first to compare deterministic Looped models with stochastic CoT.
>
> > Q1: **Assumption Robustness:** The theoretical analysis relies on several strong assumptions (e.g., polynomial-size graphs, log-precision). How sensitive are the conclusions to these assumptions? Would relaxing them change the separation results?
>
> The theory is commonly formulated under a specific set of assumptions, and it is not intended to describe regimes where these assumptions are substantially violated. This is a common limitation of most theoretical analyses: when the structural assumptions no longer hold, the analysis does not necessarily degrade smoothly, but instead becomes inapplicable. We would like to emphasize that in computational complexity theory, strong structural assumptions are often necessary in order to reason about worst-case hardness.
>
> In fact, our empirical observations indicate that the phenomenon still appears even when the assumptions are not strictly enforced.
> However, these experiments only reflect a finite set of instances and therefore cannot substitute theoretical guarantees.
> From a complexity-theoretic perspective, it is essential to analyze worst-case hardness rather than “empirically typical’’ behavior, which is why these assumptions remain necessary in our proofs.
>
> > Q2: **Empirical Validation:** The experiments (e.g., DAG and DNF counting) are highly synthetic. Can the authors demonstrate results on realistic reasoning benchmarks to confirm the claimed separation holds empirically?
>
> Since our primary contribution is theoretical, the objective of experiments is to validate our formal analysis. Such validation must rely on tasks whose computational complexity is precisely characterized, following prior theoretical work [1,2]. Realistic benchmarks, such as math problems, require composite skills and language understanding, and thus cannot cleanly validate the specific theoretical separation we aim to study. However, we agree that evaluation on realistic tasks is important and added to the conclusion as an important direction for future work.
>
> **Before:** Our findings are further substantiated by experimental evidence.
> **After:** While our results are supported by experiments on synthetic tasks whose complexity is precisely characterized, future work will investigate whether the same holds on realistic benchmarks.
>
> ---
> [1] Feng et al., Towards Revealing the Mystery behind Chain of Thought (NeurIPS 2023)
> [2] Li et al., Chain of Thought Empowers Transformers to Solve Inherently Serial Problems (ICLR 2024)

---

### Author Response · Authors · 2025-11-21
**General Response by Author**

We thank the reviewers for their thoughtful comments and for raising concerns regarding the assumptions and experimental design. We would like to clarify that these assumptions and methodologies are the established standard in this line of research at NeurIPS, ICLR, and ICML, where our paper is positioned within the formal-language-theoretic study of Transformer expressivity.

### On the theoretical assumptions:
The assumptions of logarithmic precision and polynomial-size embeddings are not only standard but necessary for formal analyses of Transformer expressivity, and they are widely adopted in prior work rather than newly introduced in our study. The polynomial-size assumption [2,4,6,7] is inherited directly from circuit complexity, where classes such as NC and TC are defined over polynomial-size circuit families. The log-precision assumption [1-7] is needed for representing positional information.

### On the synthetic experiments:
Our work is primarily theoretical, and empirical validation must rely on tasks whose computational complexity is precisely characterized. Such tasks are necessarily synthetic, and the standard evaluation methodology in prior work [1,2,4,5]. Several influential studies in this line of work include minimal or even no empirical experiments [3,6,7], reflecting the nature of the field rather than any limitation of our approach.

These assumptions and the use of synthetic tasks are not choices unique to our work; they constitute the standard methodology in formal analyses published at NeurIPS, ICLR, and ICML. Evaluating our submission negatively for adhering to these established norms would be inconsistent with prior work and could compromise the fairness of the review process.

---
[1] Feng et al., Towards Revealing the Mystery behind Chain of Thought (NeurIPS 2023)
[2] Li et al., Chain of Thought Empowers Transformers to Solve Inherently Serial Problems (ICLR 2024)
[3] Merrill et al., The Expressive Power of Transformers with Chain of Thought (ICLR 2024)
[4] Sanford et al.,Transformers, parallel computation, and logarithmic depth (ICML 2024 Spotlight)
[5] Merrill et al., A Little Depth Goes a Long Way (NeurIPS 2025)
[6] Merrill et al., Exact Expressive Power of Transformers with Padding (NeurIPS 2025)
[7] On the Reasoning Abilities of Masked Diffusion Language Models (ICLR 2026 under review)

---

### Meta-Review · Area_Chair_JJyW · 2026-01-07

**Summary:**

Reviewers have the following concerns:

1. The assumptions are a bit simplified.
2. Writing is not clear and has some logical issues. It would be great if the authors could provide more intuitions how the proof works.
3. Limited experiments with simple and synthetic dataset.

**Reviewer Concerns:**

The concerns do not seem to be addressed by the rebuttal. Reviewers do not appear to be moved. No further experiments are given by the authors.

**Reviewer Scores:**

The reviewers' scores are likely not to change.

AC has theoretical background and read the paper in details. The paper is in some sense quite similar to "H. Zhu et al, Reasoning by Superposition: A Theoretical Perspective on Chain of Continuous Thought", released in May 2025 (and accepted in NeurIPS'25). They both show that the latent thoughts are more efficient than CoTs, both construct the weights in the Transformer to achieve the goal (e.g., expressibility), and both did the experiments on parallelizable tasks. However the work does not cite that paper. Please make a detailed comparison in the next revision. Also how to empirically evaluate Sec. 4?

---

### Decision · Program_Chairs · 2026-01-26

Reject